



# Environmental sensitivities of shallow-cumulus dilution – Part 2: Vertical wind profile

Sonja Drueke[1], Daniel J. Kirshbaum[1], and Pavlos Kollias[2]

[1]Department of Atmospheric and Oceanic Sciences, McGill University, Montréal, QC, Canada
[2]School of Marine and Atmospheric Sciences, Stony Brook University, Stony Brook, NY, USA

**Correspondence:** Sonja Drueke (sonja.drueke@mail.mcgill.ca)

**Abstract.** This second part of a numerical study on shallow-cumulus dilution focuses on the sensitivity of cloud dilution to changes in the vertical wind profile. Insights are obtained through large-eddy simulations of maritime and continental cloud fields. In these simulations, the speed of the initially uniform geostrophic wind, and the strength of geostrophic vertical wind shear in the cloud and subcloud layer are varied. Increases in the cloud-layer vertical wind shear (up to 9 m s$^{-1}$ km$^{-1}$)

lead to 40–50 % larger cloud-core dilution rates compared to their respective unsheared counterparts. When the background wind speed, on the other hand, is enhanced by up to 10 m s$^{-1}$ and subcloud-layer vertical wind shear develops or is initially prescribed, the dilution rate decreases by up to 25 %. The sensitivities of the dilution rate are linked to the updraft strength and the properties of the entrained air. Increases in the wind speed or vertical wind shear result in lower vertical velocities across all sets of experiments with stronger reductions in the cloud-layer wind shear simulation (27–47 %). Weaker updrafts are exposed

to mixing with the drier surrounding air for a longer time period, allowing more entrainment to occur (i.e., the "core exposure effect"). However, reduced vertical velocities, in concert with increased cloud-layer turbulence, also assist in widening the humid shell surrounding the cloud cores, leading to entrainment of more humid air (i.e., the "core-shell dilution effect"). In the experiments with cloud-layer vertical wind shear, the core exposure effect dominates and the cloud-core dilution increases with increasing shear. Conversely, when the wind speed is increased and subcloud-layer vertical wind shear develops or is

imposed, the core-shell dilution effect dominates to induce a purifying effect. The sensitivities are generally stronger in the maritime simulations, where weaker sensible heat fluxes lead to narrower, more tilted, and, therefore, more suppressed cumuli when cloud-layer shear is imposed. Moreover, in the experiments with subcloud wind shear, the weaker baseline turbulence in the maritime case allows for a larger turbulence enhancement, resulting in a widening of the transition zones between the cores and their environment, leading to the entrainment of more humid air.

## 1 Introduction

Shallow cumuli are strongly affected by the ingestion of surrounding air, a process known as entrainment. Entrainment is caused by turbulent circulations that generate mixing along the cloud boundaries ("turbulent" entrainment) as well as cloud-scale dynamical circulations that draw organized inflow ("dynamic" entrainment) (e.g., Houghton and Cramer, 1951; de Rooy et al., 2013). Entrainment leads to the dilution of cloudy updrafts through mixing with drier and cooler air, which evaporates





cloud hydrometeors and reduces the updraft buoyancy. As a result, it tends to suppress vertical cloud development (e.g., Derbyshire et al., 2004; Gerber et al., 2008; Krueger, 2008; Del Genio, 2012; Lu et al., 2013).

Traditionally, entrainment has been conceptualized as a direct exchange of air between clouds and their undisturbed environment (e.g., Betts, 1975; Siebesma and Cuijpers, 1995; Siebesma, 1998; de Rooy et al., 2013). More recently, however, attention has turned to the importance of the thin "shell" of air surrounding the cloud in buffering the mixing process (e.g., Heus and

Jonker, 2008; Wang and Geerts, 2010; Dawe and Austin, 2011; Lamer et al., 2015; Hannah, 2017; Endo et al., 2019). The shell contains a mixture of cloud and environmental air, and thus represents a transition zone between in-cloud and environmental conditions. Importantly, air entrained from the shell causes less dilution than air entrained from the undisturbed environment.

The term "shell" has been used to describe different parts of a cumulus cloud. Heus and Jonker (2008) define the "cloud shell" as the subsiding air at the cloud edge and outside the cloud, which tends to be more humid than the surrounding environment.

In cloud simulations, Hannah (2017) referred to the "cloudy shell" as the cloudy grid points surrounding the cloud core, where the core is the positively buoyant and ascending portion of the cloud. Also, Dawe and Austin (2011) defined the "cloud-core shell" as the grid points immediately adjacent to the cloud core (whether cloudy or not). Although each definition is slightly different, they all refer to buffer zones immediately surrounding a cloud or cloud core.

The dilution experienced by shallow cumuli is partially controlled by environmental conditions. In the first part of this study,

we used LES to investigate the impacts of selected thermodynamics conditions on the cloud-core dilution (Drueke et al., 2020). The core dilution rate was found to correlate strongly, and positively, with cloud-layer relative humidity (RH), consistent with various studies (e.g., Wang and McFarquhar, 2008; Stirling and Stratton, 2012; Lu et al., 2018; Bera and Prabha, 2019). This finding can be explained by a simple buoyancy-sorting argument. Drueke et al. (2020) also found a strong sensitivity of shallow-cumulus dilution to continentality, in that simulated maritime cumuli experienced about twice the dilution of corresponding

continental cumuli. The sensitivity was linked to larger cloud-base mass fluxes over land, driven by stronger sensible heat fluxes and subcloud turbulence. Additionally, Drueke et al. (2020) found the cloud dilution to be relatively insensitive to cloud- and subcloud-layer depths. A doubling of the former resulted in only a 2–3 % change of the dilution rate and a 50 % increase in the latter resulted in only a 4 % decrease in dilution.

A consistent theme in LES cloud studies is that wider clouds tend to undergo less dilution, become more vigorous, and

undergo deeper ascent than narrower clouds (Khairoutdinov and Randall, 2006; Kirshbaum and Grant, 2012; Rieck et al., 2014; Rousseau-Rizzi et al., 2017). The concept of cloud radius ($R$) regulating cloud dilution has prevailed for decades (e.g., Morton, 1957), and can be explained by the notion that, as $R$ increases, the entrainment flux into the cloud, which depends on the cloud circumference, cannot keep pace with the increasing cloud cross-sectional area. While Drueke et al. (2020) also found a generally strong correlation between cloud width and cloud dilution, it was not universal. Thus, while $R$ is an important

controlling parameter, its effects may be overwhelmed by other factors.

Cloud vertical velocity ($w$) is also strongly related to cloud dilution. Although a robust inverse relationship between the bulk dilution rate ($\varepsilon$) and $w$ has been reported in LES (Neggers et al., 2002; Tian and Kuang, 2016; Lu et al., 2018) and observations (Kirshbaum and Lamer, 2021), the mechanisms behind this trend are unclear. From one perspective, dilution may be thought to control $w$ by reducing cloud buoyancy and mixing lower-$w$ surrounding air into the cloud. While recent LES





studies suggest that the latter "direct" effect is weak (e.g., de Roode et al., 2012; Sherwood et al., 2013; Romps and Charn, 2015), the corresponding entrainment-induced buoyancy loss remains important. From the opposite perspective, cloud dilution may be thought to depend on $w$, because $w$ determines the time scale over which clouds are exposed to environmental air (Neggers et al., 2002).

The present study focuses on the sensitivity of shallow-cumulus dilution to the geostrophic vertical wind profile. While ver-
tical wind shear is known to organize deep convection into particularly intense manifestations (e.g., supercell thunderstorms), it has more subtle effects on shallow cumuli. In principle, this shear can enhance cloud entrainment via increased turbulent mixing and/or stronger cloud-relative winds (e.g., Markowski and Richardson, 2010). Moreover, the shear tilts moist thermals downshear with height (e.g., Malkus, 1952; Asai, 1964), which enhances adverse vertical perturbation pressure gradients to weaken updraft accelerations (e.g., Parker, 2010; Peters, 2016; Helfer et al., 2020). Linear theory suggests that this shear-
induced updraft suppression depends on cloud width, with the strongest suppression for the narrowest, most vertically tilted, clouds (Kirshbaum and Straub, 2019).

Vertical wind shear also tends to displace the cloud core from the cloud center, with the maximum buoyancy, vertical velocity and liquid water content all shifting to the upshear side of the cloud (e.g., Heus and Jonker, 2008). This asymmetry is consistent with the linear theory of Rotunno and Klemp (1982), who showed that vertical shear induces a perturbation pressure dipole
across the updraft with high pressure on the upshear flank and low pressure on the downshear flank. These pressure anomalies cause the impinging flow to divert around the upshear side of the cloud and converge on the downshear side. As a result, the upshear side exhibits weakened dilution while the downshear side exhibits enhanced dilution (e.g., Heymsfield et al., 1978; Zhao and Austin, 2005). Similar to flow separation around a mountain barrier (e.g., Smolarkiewicz and Rotunno, 1989), a turbulent and moist wake also forms downshear of the cloud (e.g., Perry and Hobbs, 1996; Heus and Jonker, 2008).

Despite receiving significant attention, the impacts of vertical wind shear on $\varepsilon$ remain unclear. Both numerical simulations (Brown, 1999; Lin, 1999; Helfer et al., 2020) and observational $\varepsilon$ retrievals (Kirshbaum and Lamer, 2021) suggest minimal sensitivity of $\varepsilon$ to cloud-layer shear. However, these findings counter the logic of Neggers et al. (2002) that weaker updrafts (here, due to shear-enhanced vertical perturbation pressure gradients) should enhance cloud dilution. To resolve this apparent contradiction, more detailed analyses of the impacts of vertical wind shear on shallow cumuli are needed. Furthermore, little
attention has been paid to the general impact of background winds on shallow-cumulus dilution. Although a uniform background flow does not directly impact cumuli, it may indirectly affect them by modifying the subcloud flow. As the background winds increase, so does the frictionally induced vertical shear in the subcloud layer, which can extend into the cloud layer and/or organize the subcloud turbulence into shear-parallel rolls (e.g., Weckwerth et al., 1997). The latter are associated with elongated updrafts in the shear direction that, upon reaching saturation at cloud base, may give rise to larger and less dilute
cumuli.

While no studies to our knowledge have directly investigated the relationship between background winds and cloud dilution, some offer insights into how simulated clouds may respond to increased wind speeds. Nuijens and Stevens (2012) found a positive correlation between background wind speed and cloud depth in simulated trade-wind cumuli, an effect that may have been accompanied by decreased cloud dilution. Also, from a purely numerical perspective, the degree of model diffusion is





sensitive to cross-grid wind speed. Cloud models typically use highly diffusive flux-limited, flux-corrected, and/or monotonic advection schemes to damp spurious small-scale oscillations generated by advective errors near cloud surfaces. In the presence of a cross-grid flow, these schemes tend to produce enhanced diffusion in the flow direction, which can spuriously enhance cloud size and thereby weaken cloud dilution (Wyant et al., 2018).

To study the impacts of the vertical wind profile on shallow-cumulus dilution, we conduct LES of shallow-cumulus en-
sembles in which aspects of this wind profile are systematically varied. The model configuration is provided in Sect. 2, and the experimental results are presented in Sect. 3. Sect. 4 provides a physical explanation of the various sensitivities of cloud dilution, and proposes a new empirical formulation for the dilution rate. Section 5 provides the conclusions.

## 2 Methodology

### 2.1 Model configuration

As in Part I of this study, we conduct LES of shallow cumulus ensembles using the Bryan Cloud Model version 17 (CM1; Bryan and Fritsch, 2002). In LES mode, CM1 accurately reproduces the findings from past LES inter-comparison studies of shallow cumuli (Drueke et al., 2019, 2020). To examine diverse cloud fields, we consider one maritime case and one continental case, the former based on the LES inter-comparison study of the Barbados Oceanographic and Meteorological Experiment (BOMEX) by Siebesma et al. (2003), and the latter based on the LES inter-comparison of shallow cumuli at the
US Atmospheric Radiation Measurement (ARM) Southern Great Plains (SGP) observatory in Oklahoma (Brown et al., 2002).

The model configuration is similar to that in Drueke et al. (2020), with a monotonic fifth-order weighted essentially non-oscillatory (WENO) advection scheme for both scalars and velocity, a third-order Runge-Kutta time-differencing scheme, and periodic horizontal, semi-slip lower, and free-slip upper boundary conditions as well as a $f$-plane approximation. The Coriolis force is applied to wind perturbations from the initial, geostrophic profile. A horizontal grid spacing of 32 m is
used to adequately resolve the turbulent circulations of interest. The BOMEX horizontal domain size of $6.4\times6.4$ km$^2$ is left unchanged from Siebesma et al. (2003), while the ARM-SGP domain size is doubled from $6.4\times6.4$ km$^2$ in Brown et al. (2002) to $12.8\times12.8$ km$^2$ to capture the larger-scale circulations of this cloud field. For each experiment, an ensemble of six members is conducted, each with a different field of small-amplitude random perturbations added to the initial potential temperature and water-vapor mixing ratio fields. The results presented for each case are averaged over this ensemble.

### 2.2 LES experiments

We conduct various idealized experiments to quantify the impacts of the initial wind profile on the cloud dilution. To examine the impacts of cloud-layer vertical shear, the first set of experiments (CL-SHR) initializes zero wind in the subcloud layer and positive, linear westerly vertical shear in the cloud layer (Fig. 1). While the absence of subcloud winds differs from the standard configurations of these cases, it limits the development of subcloud vertical shear that, as will be seen, may indirectly
affect cloud dilution. To ensure that the shear layer is fully contained within the cloud layer, the shear base is placed at 720 m


in BOMEX and 1000 m in ARM-SGP (dashed lines in Fig. 1). Zonal vertical shears ranging from 0 to 9 m s$^{-1}$ km$^{-1}$ (CTRL to S9), in increments of 3 m s$^{-1}$ km$^{-1}$, are applied from the shear base to the domain top. For BOMEX, we also include an experiment with vertical wind shear of 1.8 m s$^{-1}$ km$^{-1}$, matching that of Siebesma et al. (2003) (Table 1).

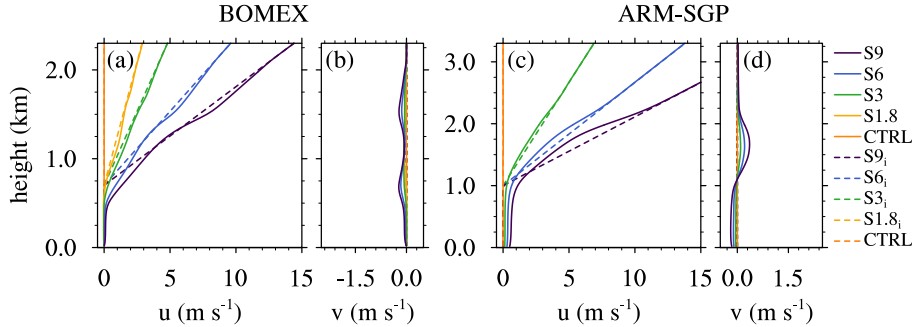

**Figure 1.** Initial wind profiles (dashed lines) and wind profiles averaged over the analysis period (solid lines) for the CL-SHR (**a-b**) BOMEX (3-6 h) and (**c-d**) ARM-SGP (14:00-15:00 LST) experiments.

**Table 1.** Summary of CL-SHR experiments. See text for further details.

|       | BOMEX | ARM-SGP |
|-------|:-----:|:-------:|
| CTRL  | x     | x       |
| S1.8  | x     | -       |
| S3    | x     | x       |
| S6    | x     | x       |
| S9    | x     | x       |

In a second set of experiments (WIND), we evaluate the sensitivity of simulated cloud dilution to vertically uniform zonal geostrophic winds of magnitude $U$. In line with the prevailing wind directions at the two locations, we consider easterly winds in BOMEX and westerly winds in ARM-SGP (Table 2). The winds increase from zero (CTRL) up to 10 m s$^{-1}$ (U10; dashed lines in Figs. 2a-d). Finally, to examine the impacts of subcloud geostrophic vertical shear on cloud dilution, a third suite of experiments vary the near-surface shear (SCL-SHR). These profiles are identical to those in WIND except for having zero surface wind and a layer of linear zonal shear over the lowest 250 m (dashed lines in Fig. 2e and g). Individual simulations from this suite of experiments are named based on their shear magnitude; for example, the case of 40 m s$^{-1}$ km$^{-1}$ of near-surface shear is named US40.

Due to the short durations of active cloud development ($< 6$ h) in the BOMEX and ARM-SGP simulations, we do not apply any forcings to maintain the wind profile at its initial values. The wind profiles thus vary with time, mainly through the action of subcloud and cloud-layer vertical mixing. Nevertheless, as will be seen, the qualitative differences between the various cases are maintained throughout the simulations, although slightly reduced over time (see solid lines in Figs. 1-2). To determine whether sensitivities to cross-grid flow like those highlighted by Wyant et al. (2018) affect our model results, we





**Table 2.** Summary of the WIND and SCL-SHR experiments. See text for further details.

| | BOMEX | ARM-SGP |
|---|---|---|
| **WIND (m s$^{-1}$)** | | |
| CTRL | 0.0 | 0.0 |
| U2.5 | -2.5 | 2.5 |
| U5 | -5.0 | 5.0 |
| U10 | -10.0 | 10.0 |
| **SCL-SHR (m s$^{-1}$ km$^{-1}$)** | | |
| CTRL | 0.0 | 0.0 |
| US10 | -10.0 | 10.0 |
| US20 | -20.0 | 20.0 |
| US40 | -40.0 | 40.0 |

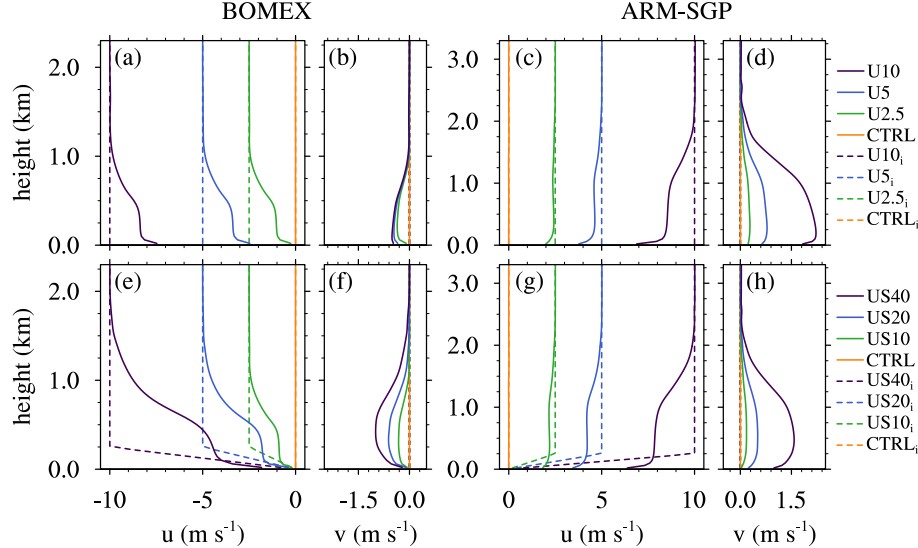

**Figure 2.** Initial wind profiles (dashes lines) and wind profiles averaged over the analysis period (solid lines) for (**a-b**) the BOMEX and (**c-b**) the ARM-SGP WIND experiments and (**e-f**) the BOMEX and (**g-h**) the ARM-SGP SCL-SHR experiments.





have compared various runs with a fixed and a translating grid (at the approximate average speed of the cloud-layer flow). The differences between these runs were minimal, suggesting that such effects are not significant for our model configuration. Therefore, for consistency, all simulations described herein use a stationary grid.

## 3   Results overview

### 3.1   The CTRL cases

The maritime BOMEX CTRL case represents a typical trade-wind cloud field, except for the lack of ambient winds. The surface heat fluxes, large-scale advection and subsidence tendencies, and simulated convection come into balance to yield a statistically quasi-steady flow over 3-6 h (Siebesma et al., 2003). The cloud base stays at roughly 500 m throughout the simulation, and the cloud top extends above the base of the trade-wind inversion at 1.5 km (Fig. 1 of Drueke et al., 2020). A horizontal cross-section of $w$ at the midpoint of the subcloud layer and a time of 4 h shows a cellular turbulence pattern with variations on a broadly similar scale ($\sim 500$ m) as the subcloud layer depth (Fig. 3a). The small-scale subcloud turbulence gives rise to small active cumuli with mean radii at the level of free convection ($R_{\mathrm{LFC}}$) of 80 m (Fig. 3c). Active clouds are defined as clouds possessing a positively buoyant and ascending internal core, and a circular cloud shape is used to infer $R_{\mathrm{LFC}}$ based on the horizontal area occupied by the cloud.

The continental ARM-SGP CTRL case, in contrast, exhibits a time-evolving cloud field forced by the diurnal cycle of the surface heat fluxes (Brown et al., 2002). Shallow cumuli first initiate at about 11:00 local solar time (LST) and dissipate by around 20:00 LST. Over that time, the cloud base rises from 0.6 km to 1.3 km (Drueke et al., 2020). As shown by the $w$ cross-section at the midpoint of the subcloud layer at 13:15 LST (Fig. 3b), the deeper subcloud layer in ARM-SGP gives rise to larger horizontal circulations than in BOMEX. As a result, $R_{\mathrm{LFC}}$ in ARM-SGP is larger (Fig. 3d), with an averaged value (207 m) more than double that of BOMEX.

Based on the time-evolution of the BOMEX and ARM-SGP simulations, we define analysis periods to be used for the detailed calculations to follow. These periods are selected to avoid model spin-up or cloudless intervals, thus focusing on the well developed turbulent cloud fields of interest. The quasi-stationarity of the BOMEX case permits the use of a relatively long 3-h averaging period, covering 3-6 h. Due to the diurnal evolution of the cloud field in the continental ARM-SGP experiments, a shorter averaging time of one hour is used, running from 14:00-15:00 LST. Unless otherwise specified, all calculations herein are conducted during these analysis periods.

### 3.2   Sensitivity to cloud-layer vertical wind shear

Over the course of the BOMEX CL-SHR simulations, the shear base lowers from its initial value (720 m) down to about 500 m due to cloud-layer vertical mixing. As a result, the cloud base and shear base nearly coincide over the analysis period (solid lines in Figs. 1a-b). The flow remains predominately westerly with a weak northerly component in the cloud layer. Similarly, in the ARM-SGP CL-SHR simulations, the shear base over the analysis period roughly coincides with the cloud

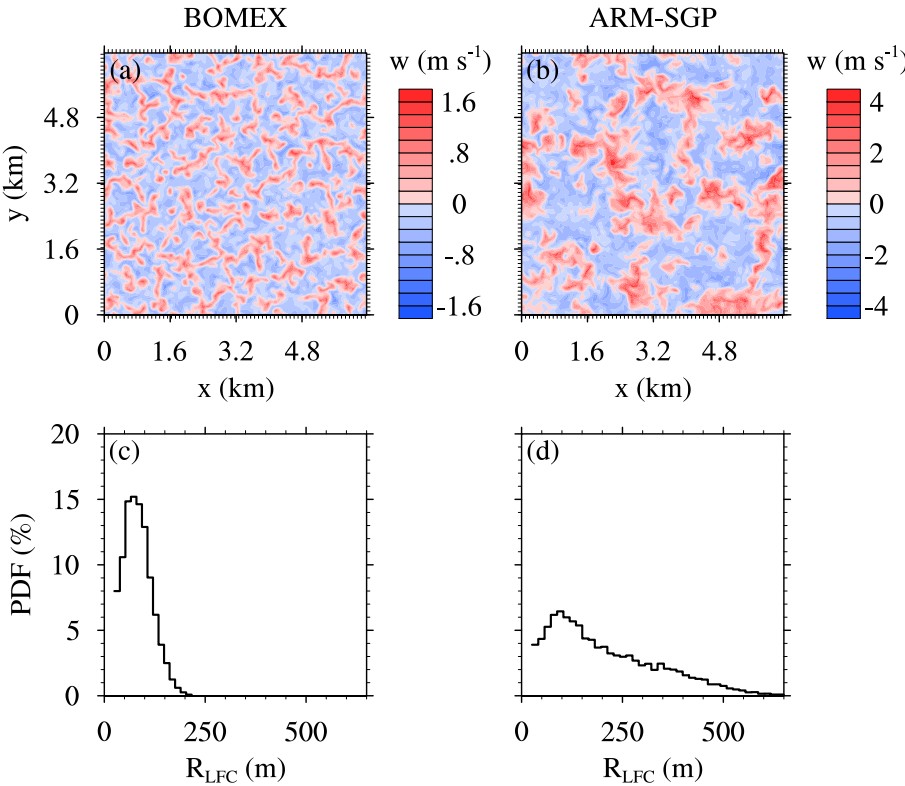

**Figure 3.** Instantaneous cross-section of the vertical velocity at the midpoint of the subcloud layer of the CTRL experiments in (**a**) BOMEX at 4 h and (**b**) ARM-SGP at 13:15 LST. For ARM-SGP, a subsection of equal size to the BOMEX domain is shown. (**c-d**) The probability density function (PDF) of the cloud radius at LFC of all active clouds. Panels (**a**) and (**c**) show the maritime BOMEX case, and the panels (**b**) and (**d**) the continental ARM-SGP case.

base at 1050 m (Figs. 1c-d). Although the subcloud winds remain weak, they turn more northerly due to larger surface drag, enhanced turbulent mixing, and a stronger Coriolis force. In both cases, the cloud-layer shear weakens modestly over time

(Figs. 1a and c). In the most extreme S9 case, the cloud-mass-flux-weighted shear weakens to 6.4 m s$^{-1}$ km$^{-1}$ (BOMEX) and 6.9 m s$^{-1}$ km$^{-1}$ (ARM-SGP) over the analysis period, a reduction of around 25 %. Thus, despite the gradually weakening shear, the two sets of simulations exhibit comparable shear magnitudes throughout.

As the cloud-layer shear is increased, the subcloud layer is minimally affected, with similar $w$ fields in the CTRL and S9 cases (cf. Figs. 3a-d and Figs. 4a-b). In the cloud layer, the clouds widen with increasing shear, a signal that extends down to

the LFC. The $R_{\mathrm{LFC}}$ distributions shift toward larger scales in both BOMEX and ARM-SGP (Figs. 4c-d), but much more so in BOMEX. The mean $R_{\mathrm{LFC}}$ increases by 27 % in BOMEX, compared to 14 % in ARM-SGP.

Although not shown for brevity, the simulated cumuli respond to the imposed vertical shear in expected ways: (i) they tilt downshear with height (e.g., Malkus, 1952; Asai, 1964), (ii) they develop pressure-anomaly dipoles straddling the clouds along the shear axis, with high pressure upshear and low pressure downshear (e.g., Rotunno and Klemp, 1982; Zhao and



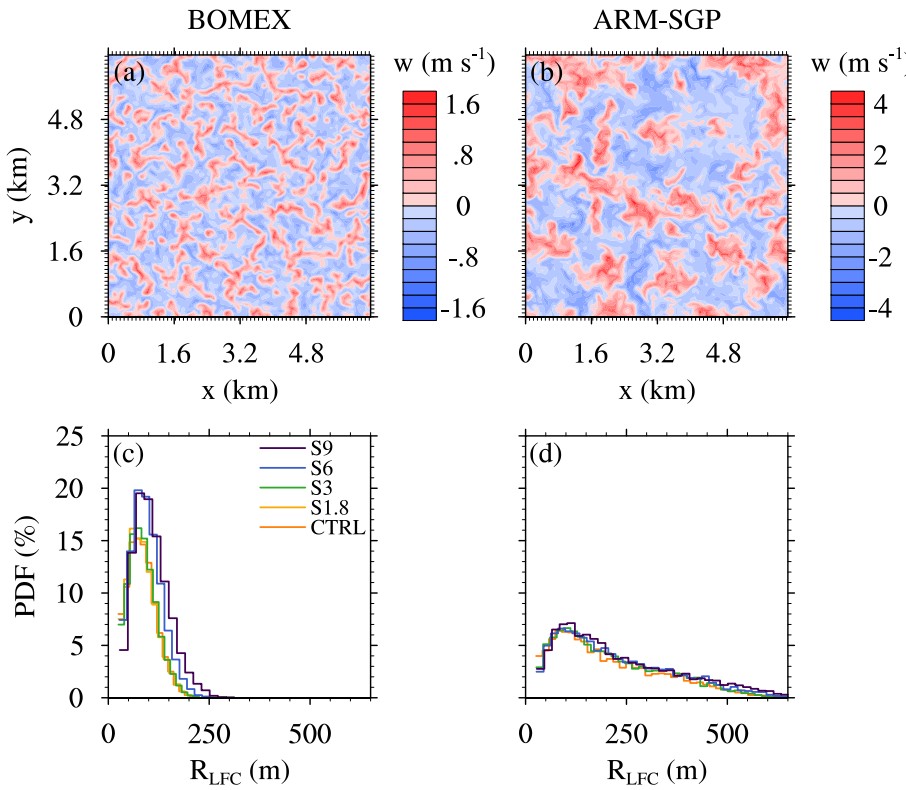

**Figure 4.** As in Fig. 3 but for the CL-SHR experiments with panel (**a**) and (**b**) showing the horizontal cross-section of the vertical velocity at the subcloud-layer midpoint of the S9 experiments.

Austin, 2005), and (iii) their buoyant cloud cores shift to the upshear side of the cloud (e.g., Heymsfield et al., 1978; Heus and Jonker, 2008). Moreover, the cloud-top heights decrease under stronger vertical shear, suggesting a shear-induced cloud suppression. Compared to the CTRL experiments, the cloud-top height decreases by 15–20 % in the S9 versions of the BOMEX and ARM-SGP cases (not shown).

    Diagnosis of the simulated bulk fractional entrainment rate, or simply the "dilution rate", follows the formulation of

Siebesma and Cuijpers (1995, hereafter SC95). Specifically, we infer dilution based on the budget of the total water specific humidity ($s_\mathrm{t}$), which includes the vertical gradient of $(s_\mathrm{t})_\mathrm{co}$, the turbulent fluxes, time derivative of $(s_\mathrm{t})_\mathrm{co}$, and large-scale forcings, where "co" denotes conditional averages within cloud cores. To obtain the bulk dilution rate ($\varepsilon$), we divide the total bulk entrainment rate by the core mass flux ($M_\mathrm{co} = \rho a_\mathrm{co} w_\mathrm{co}$), where $\rho$ is the air density, $a_\mathrm{co}$ the cloud-core fraction, and $w_\mathrm{co}$ the conditionally averaged cloud-core vertical velocity. Instantaneous vertical profiles of $\varepsilon$ are calculated at each model output

time over the full horizontal domain, at all vertical levels where cloud core grid points are found. To compare $\varepsilon$ across the different simulations, we perform some averaging to obtain representative values. First, 15-min running averages are calculated from the 5-min model output data, from which bulk cloud-layer averages are computed over the central 50 % of the cloud layer. Both the vertical profiles and the bulk values are then averaged over the analysis period.




The $\varepsilon$ profiles thus obtained increase monotonically with vertical wind shear (Fig. 5). This sensitivity originates at cloud
base, increases to a maximum near the cloud-layer midpoint, and decreases rapidly near cloud top. Averaged over the central
50 % of the cloud layer, $\varepsilon$ in the BOMEX S9 experiment (2.6 km$^{-1}$) is about 50 % larger than in the CTRL simulation
(1.7 km$^{-1}$). This sensitivity is comparable but slightly weaker ($\sim 40$ %) in ARM-SGP.

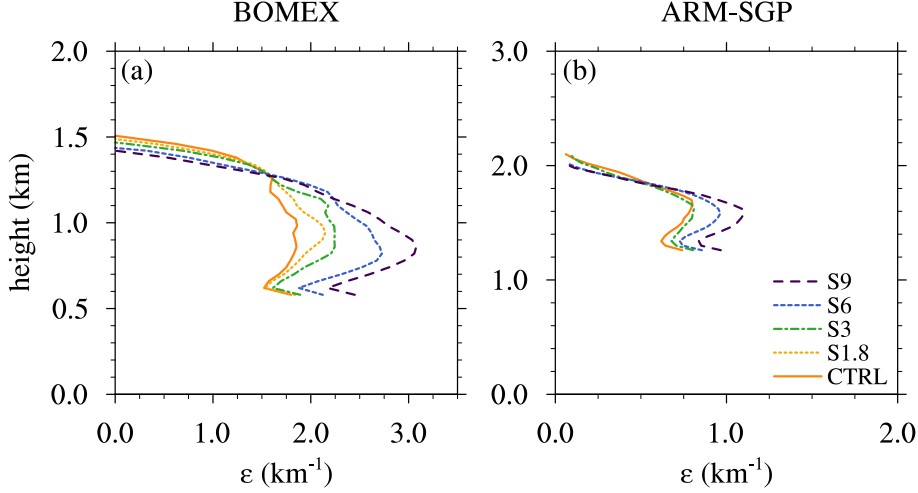

**Figure 5.** Dilution rate ($\varepsilon$) for the CL-SHR experiments for **(a)** BOMEX and **(b)** ARM-SGP.

### 3.3 Sensitivity to vertically uniform geostrophic winds

The initially uniform zonal velocity profiles from the WIND simulations evolve into vertically varying profiles with strong low-
level vertical shear. At analysis time, the zonal wind is strongly forward-sheared near the surface, exhibits a nearly constant
value within the central part of the subcloud layer, and then exhibits additional forward shear extending into the lower cloud
layer (Figs. 2a, c). The Coriolis force induces a cyclonic turning of these frictionally decelerated winds, leading to a meridional
wind component that is maximized within the subcloud layer (Figs. 2b, d). As with the CL-SHR experiments, the stronger
surface drag and Coriolis force in ARM-SGP leads to stronger frictional deceleration and cyclonic wind turning than in the
corresponding BOMEX simulations.

In BOMEX U10, the low-level shear organizes the subcloud flow into longitudinal bands, or rolls, aligned with the low-level
winds (Fig. 6a), which contrasts with the more disorganized convection in ARM-SGP U10 (Fig. 6b). These differing responses
likely stem from the different surface heating rates in the two cases. The larger surface buoyancy flux in ARM-SGP yields a
more buoyancy-dominated convective boundary layer (CBL), which overwhelms any shear-induced turbulence organization.
The potential for convective rolls may be assessed based on the Monin-Obukhov length ($L$), where $L$ represents the height at
which buoyancy dominates over shear in the production of TKE:

$$L = -\frac{\overline{\theta}_v \left( \overline{(u'w')}^2 + \overline{(u'w')}^2 \right)^{0.75}}{kg\overline{w'\theta'_v}}, \tag{1}$$





$k = 0.4$ is the von-Kármán constant, $g$ is the gravitational acceleration, $\theta_v$ is the virtual potential temperature, $u$, $v$, and $w$ are $x$, $y$, and $z$ wind components, primes denote perturbations from a temporal or spatial average (denoted by overbars), and all

quantities are evaluated at the surface (e.g., Stull, 1985). Negative $L$ corresponds to CBLs, with smaller magnitudes reflecting more dominant buoyancy production.

Taking $z_i$ as the subcloud-layer depth and evaluating $L$ at the surface, and summing resolved and subgrid fluxes in Eq. (1), we obtain $-z_i/L = 1.6$ and $-z_i/L = 9.5$ for the BOMEX and ARM-SGP U10 cases, respectively. The smaller BOMEX value is more favorable for rolls, falling into the $0 \leq -z_i/L \leq 4.5$ range reported by Deardorff (1972) as conducive for roll

development. This roll organization at larger $U$ in BOMEX is associated with increased subcloud length scales: the wavelength of the spectral peak in the subcloud kinetic energy spectrum increases by about 110 % from CTRL to U10 (Fig. 6c) as the mean $R_{LFC}$ increases by $\sim 70$ % (Fig. 6e). For ARM-SGP, the subcloud length scales also increase, but by a much smaller amount (20 %; Fig. 6d), and $R_{LFC}$ increases minimally (Fig. 6f).

The BOMEX $\varepsilon$ decreases with increasing $U$ over most of the cloud layer, except near cloud base (Fig. 7a). Averaged over

the central 50 % of the cloud layer, $\varepsilon$ decreases by 25 % from CTRL to U10. In contrast, the ARM-SGP $\varepsilon$ changes minimally across the experiments (Fig. 7b), with the cloud-layer-averaged $\varepsilon$ decreasing by only 7 % from CTRL to U10. In both BOMEX and ARM-SGP, the $\varepsilon$ sensitivity depends on height, with the dominant trend lying in the mid-to-upper cloud layer and non-systematic variations in the lower cloud layer.

### 3.4 Sensitivity to subcloud-layer shear

The only initial difference between the WIND and SCL-SHR experiments is the latter's geostrophic subcloud wind shear. Although vertical mixing modifies the wind profiles over time, they maintain stronger vertical wind shear in the subcloud and lower cloud layer than the corresponding WIND cases (Figs. 2e-h). The strengthened cloud-layer shear is most pronounced in the BOMEX US40 case, where the shear extends up to 1.5 km, compared to only 1 km in the BOMEX U10 case. As in the WIND experiments, the subcloud shear tends to organize the subcloud turbulence into shear-parallel rolls in BOMEX (but not

in ARM-SGP), with even larger increases in the mean of the cloud-size distribution (not shown). Again, $\varepsilon$ generally decreases with increasing winds, with a larger cloud-layer-averaged decrease in BOMEX (22 %) than in ARM-SGP (13 %) between the CTRL and US40 cases (Figs. 7c-d).

The $\varepsilon$ sensitivity in BOMEX is characterized by a positive trend over 0.5-0.8 km that reverses to a negative trend above (Fig. 7c). While this feature suggests enhanced cloud dilution in the lower cloud layer, it may also point to a different effect:

elevated cloud initiation by vertically propagating internal gravity waves. As previously noted, the SCL-SHR flows exhibit wider subcloud updrafts and sharper cloud-base shears as $U$ is increased (Fig. 2e). Both factors favor vertically propagating waves via the "obstacle effect" (e.g., Gibert et al., 2011), in which wave disturbances are forced by airflow over clouds penetrating into the cloud layer. Compared to CTRL, the US40 case exhibits a much wider distribution of cloud-base heights over 0.5-0.8 km (not shown). Such heterogeneity in cloud-base height complicates the interpretation of $\varepsilon$, for reasons outlined in

Kirshbaum (2020). Namely, vertical variations in conditionally averaged core conserved properties cannot be unambiguously

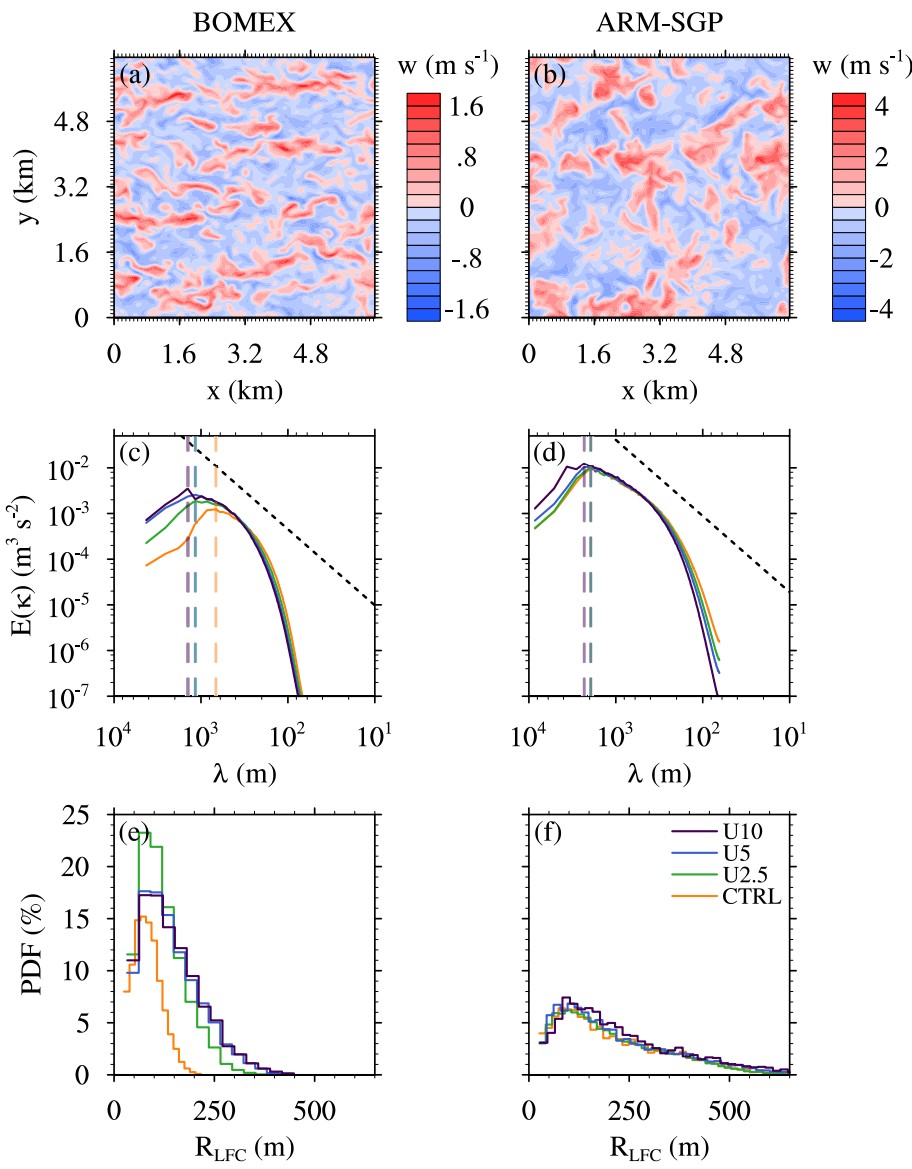

**Figure 6.** Similar to in Fig. 3 but for the WIND experiments with panel (**a**) and (**b**) showing horizontal cross-section of the vertical velocity halfway into the respective subcloud layer of the U10 experiments. (**c-d**) Two-dimensional kinetic-energy spectra in the subcloud layer for the same experiments. The black dashed lines shows the slope of $\kappa^{-5/3}$ and the colored dashed lines indicate the scales of maximum energy of the respective spectra. (**e-f**) The probability density function (PDF) of the cloud radius at LFC.





attributed to cloud dilution, because they can also be explained by variations in the source layers of different clouds. Thus, the trend in $\varepsilon$ over 0.5-0.8 km may be more reflective of variations in cloud-base height than to variations in cloud dilution.

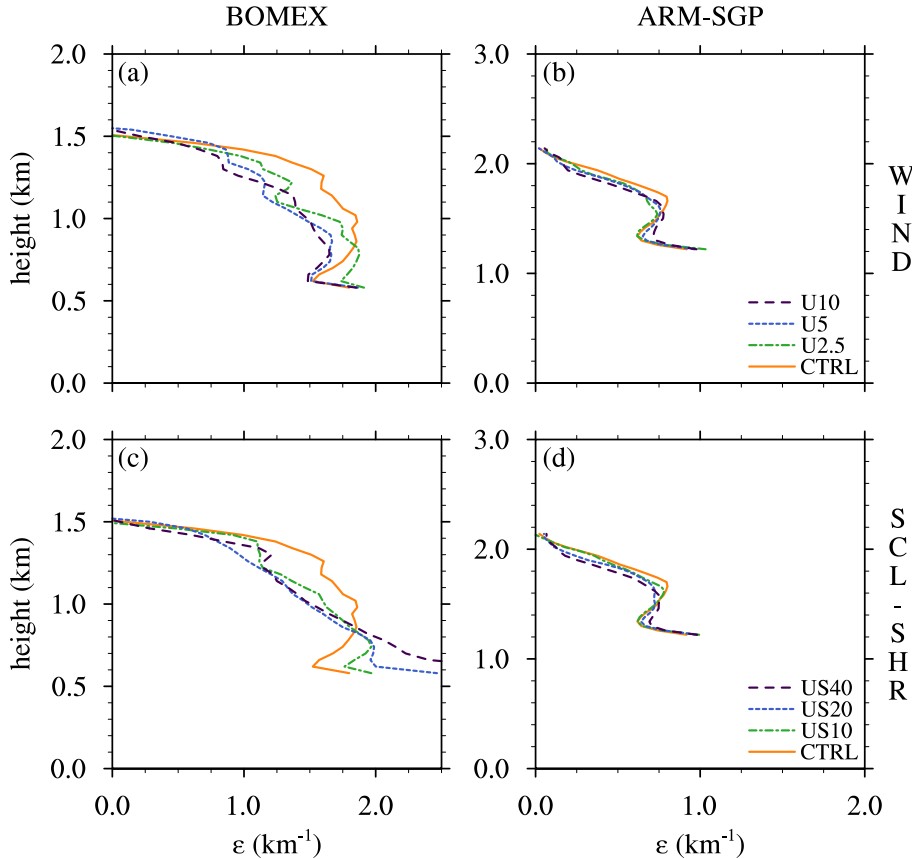

**Figure 7.** Dilution rate ($\varepsilon$) for the WIND experiments for (**a**) BOMEX and (**b**) ARM-SGP. (**c-d**) show the dilution rate profiles for the respective SCL-SHR experiments. Panels (**a**) and (**c**) are for BOMEX and panels (**b**) and (**d**) are for ARM-SGP.

# 4 Physical interpretation

Two factors have been found to jointly explain the sensitivities of $\varepsilon$ to the initial wind profile: the cloud-core $w$ and the mixing 
fraction ($\mu$; the fraction of core air within the cloud-core shell). In the following, we investigate each factor in detail.

## 4.1 Cloud-core vertical velocity

As mentioned in Sect. 1, the conditionally averaged cloud-core $w$ (or $w_{co}$) may influence $\varepsilon$ by controlling the time scale over which ascending clouds or cloud-cores are exposed to environmental air (e.g., Neggers et al., 2002). Although such a one-way causal sensitivity between $w_{co}$ and $\varepsilon$ likely oversimplifies their relationship, the correlation between $w_{co}$ and $\varepsilon$ still merits



examination. In the CL-SHR experiments, $w_{co}$ at cloud base is larger in ARM-SGP (2.6 m s$^{-1}$) than in BOMEX (1.7 m s$^{-1}$) (Figs. 8a-b). After a brief decrease between cloud base and the LFC, $w_{co}$ rebounds to maxima near cloud top of 2.5-5 m s$^{-1}$ in ARM-SGP and 1-2.5 m s$^{-1}$ in BOMEX. The larger $w_{co}$ in ARM-SGP is owing to stronger surface heating, which drives stronger subcloud turbulence and cloud-base updrafts, in conjunction with larger cloud-core buoyancy $b_{co}$ (Figs. 9b and d), which enhances vertical motions above the LFC.

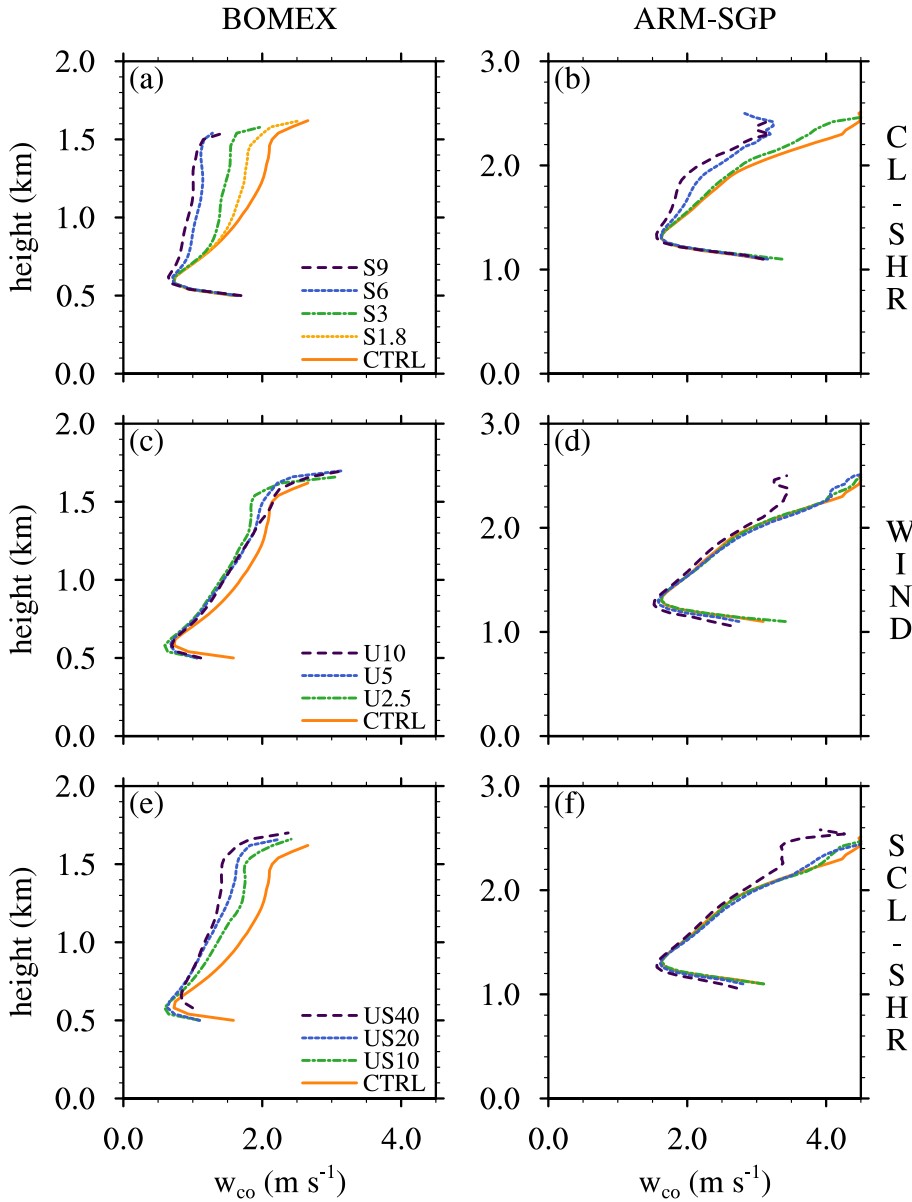

**Figure 8.** Vertical profiles of conditionally averaged cloud-core vertical velocities ($w_{co}$) for (**a-b**) the CL-SHR, (**c-d**) the WIND, and (**e-f**) the SCL-SHR experiments. Panels (**a**), (**c**), and (**e**) show the BOMEX case, and the panels (**b**), (**d**), and (**f**) the ARM-SGP one.





Cloud-layer shear induces a systematic reduction in $w_{co}$ in the CL-SHR experiments, which is expected given the tendency of this shear to tilt and weaken cumulus updrafts (e.g., Peters, 2016; Peters et al., 2019; Helfer et al., 2020). Near cloud base, where $w_{co}$ is dominated by subcloud momentum, the differences between the various cases are small. These differences increase with height to a maximum near the cloud tops. Averaged over the central 50 % of the cloud layer, BOMEX exhibits a 47 % decrease in $w_{co}$ between CTRL and S9, compared to a 27 % decrease in ARM-SGP. Hence, increased cloud-layer shear

is associated with decreased $w_{co}$ and larger $\varepsilon$, consistent with the findings of Neggers et al. (2002).

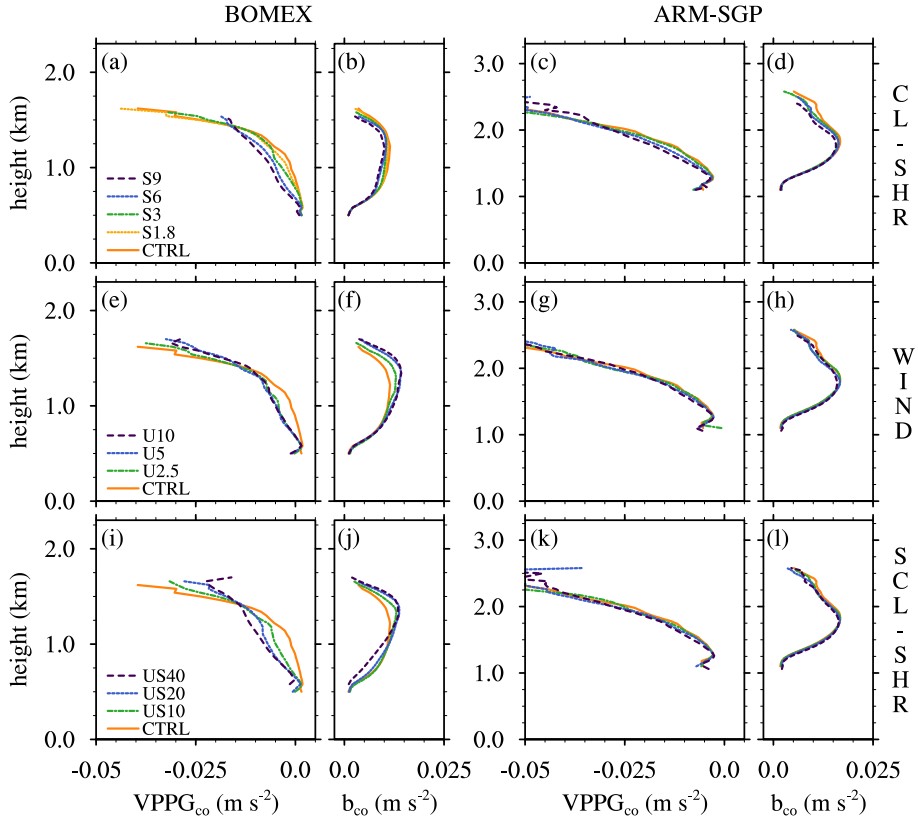

**Figure 9.** Vertical perturbation pressure gradients ($VPPG_{co}$) and conditionally averaged cloud-core buoyancy ($b_{co}$) for (**a-d**) the CL-SHR, (**e-h**) the WIND, and (**i-l**) the SCL-SHR experiments. Panels (**a-b**), (**e-f**), and (**i-j**) show the BOMEX case, and the panels (**c-d**), (**g-h**), and (**k-l**) the ARM-SGP one.

For the BOMEX WIND and SCL-SHR experiments, $w_{co}$ is again largest in the CTRL cases and decreases with increasing $U$ (Figs. 8c, e). Cloud-layer averages of $w_{co}$ decrease by 8 % and 27 %, respectively, between the CTRL and the end members of each suite (U10 and US40). By contrast, the corresponding ARM-SGP experiments are nearly insensitive to $U$, with only a 7 % and 6 % decrease in cloud-layer-averaged $w_{co}$. Unlike the CL-SHR experiments where $w_{co}$ and $\varepsilon$ varied inversely, $\varepsilon$

*decreases* with decreasing $w_{co}$ in these experiments. Thus, $w_{co}$ cannot be the only factor regulating the simulated dilution rate. Although the plot of $w_{co}^{-1}$ against $\varepsilon$ for all simulations indicates a large correlation coefficient ($R = 0.93$), the differing trends



of the CL-SHR and WIND/SCL-SHR experiments are obvious (Fig. 10a). All continental experiments are more 'resilient' to subcloud- and cloud-layer wind shear and show weaker sensitivities to the imposed changes in geostrophic winds, particularly in the WIND and SCL-SHR experiments. When only the maritime experiments are considered, the correlation coefficient

between $\varepsilon$ and $w_{\mathrm{co}}$ is substantially reduced ($R = 0.65$). Thus, while $w_{\mathrm{co}}$ strongly influences (and/or is influenced by) $\varepsilon$, other factor(s) must also be important.

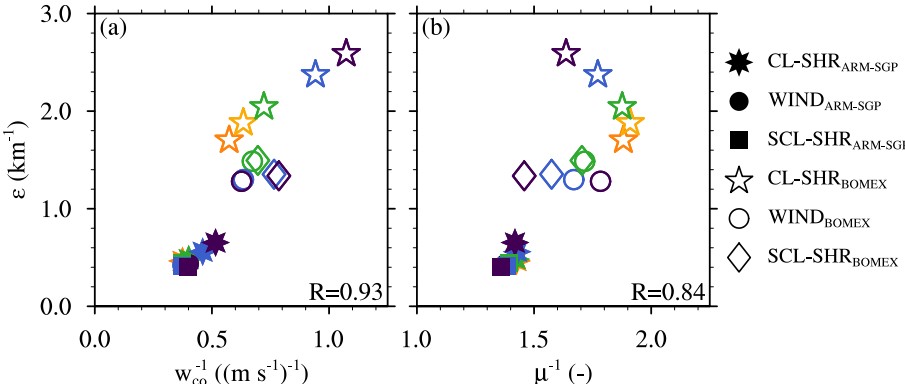

**Figure 10.** Variation of simulated dilution rate ($\varepsilon$) with (**a**) conditionally averaged core vertical velocity ($w_{\mathrm{co}}$) and (**b**) cloud-core-shell mixing fraction $\mu$, both averaged over the central 50 % of the cloud layer, for all experiments conducted herein. The color scheme is identical to Fig. 9, and the correlation coefficients for each plotted relation are shown in the lower-right corner of each plot.

To investigate the processes regulating $w_{\mathrm{co}}$, we use the core-averaged $w$ equation in CM1 (following de Roode et al., 2012):

$$\left[\frac{Dw}{Dt}\right]_{\mathrm{co}} = -\left[c_{\mathrm{p}}\theta_{\rho}\frac{\partial \pi'}{\partial z}\right]_{\mathrm{co}} + b_{\mathrm{co}} - \frac{\varepsilon_{\mathrm{w}}w_{\mathrm{co}}^2}{1 - a_{\mathrm{co}}}, \tag{2}$$

where $c_{\mathrm{p}}$ is the specific heat of dry air at constant pressure, $\theta_{\rho}$ is the density potential temperature, $\pi$ is the Exner function and $\pi'$ its perturbation relative to the horizontal average, $\varepsilon_{\mathrm{w}}$ is the fractional entrainment rate of $w$, and the effects of subgrid turbulent mixing are neglected. The dominant terms on the right side of Eq. (2) are the first two (pressure gradient and buoyancy) (e.g., Tang and Kirshbaum, 2020), and we henceforth neglect the entrainment term because it has been found to be small (e.g., de Roode et al., 2012).

The expected tendency for cloud-layer shear to enhance the adverse vertical perturbation pressure gradient (or VPPG$_{\mathrm{co}}$) is reproduced in the CL-SHR experiments, but more strongly so in BOMEX than in ARM-SGP (Figs. 9a, c). To interpret why the BOMEX VPPG$_{\mathrm{co}}$ is more sensitive to the shear, we use the linear theory of shallow convection in Kirshbaum and Straub (2019), who found that the VPPG-induced updraft suppression depends on the cloud width and layer depth (Appendix A). Because narrower and taller clouds are more tilted by the shear than are wider, shallower clouds, they experience a larger VPPG$_{\mathrm{co}}$

enhancement with increasing shear. The convective growth rate ($\sigma$) calculated using the linear theory for both BOMEX and ARM-SGP is more than halved between the CTRL and S9 cases. The marginal reduction owing to the shear may be measured by the ratio of the growth rates in the S9 and CTRL cases, which is smaller for BOMEX (0.33) than for ARM-SGP (0.45),





suggesting greater shear-induced suppression for the narrower clouds in BOMEX. Although these differences in $\sigma$ are not dramatic, they lead to large differences over time because $\sigma$ is an exponential argument. For example, over a 10-min period

representing the growing phase of a shallow cumulus (e.g., Rauber et al., 2007), the theoretical shear-induced reduction in $w$ becomes twice as large in BOMEX as in ARM-SGP.

The second important term in Eq. (2) is the cloud-core buoyancy, which is highly sensitive to lateral entrainment (e.g., Kirshbaum and Grant, 2012). Neggers et al. (2002) argued that a faster ascending core experiences less entrainment and, hence, maintains larger buoyancy, which further accelerates its ascent. Although $b_{\rm co}$ and $w_{\rm co}$ both decrease with increasing

vertical shear in the CL-SHR experiments, the $b_{\rm co}$ sensitivity is comparatively modest and of similar strength in BOMEX and ARM-SGP (Figs. 9b, d). Thus, while both of the dominant terms in Eq. (2) tend to suppress $w_{\rm co}$ in shear flows, the VPPG$_{\rm co}$ term largely explains the contrasting sensitivities of $w_{\rm co}$ to cloud-layer shear in BOMEX and ARM-SGP (Figs. 8a-b).

The adverse VPPG$_{\rm co}$ also strengthens with increasing $U$ across the BOMEX WIND and SCL-SHR experiments, at a magnitude comparable to that in the CL-SHR experiments (Figs. 9e, i). This result contrasts sharply with the minimal corresponding

variations in ARM-SGP (Figs. 9g, k). Given that the VPPG$_{\rm co}$ sensitivity in CL-SHR was attributed to the prescribed cloud-layer shear, it is fair to wonder if the VPPG$_{\rm co}$ sensitivity in BOMEX is owing to the strong lower-cloud-layer shear that develops in the WIND and SCL-SHR suites (Figs. 2a-d). While this shear likely plays an important role in enhancing the VPPG$_{\rm co}$ in the lower cloud layer, it gradually decays with height above cloud base. However, the VPPG$_{\rm co}$ sensitivity extends throughout the cloud layer, suggesting that the lower-cloud-layer shear is not the sole cause.

Another mechanism behind the VPPG$_{\rm co}$ sensitivities in the BOMEX WIND and SCL-SHR experiments is the associated sensitivity of $b_{\rm co}$ to the background winds (Figs. 9f, j). In both sets of experiments, the maximum $b_{\rm co}$ increases with $U$. To relate this sensitivity to VPPG$_{\rm co}$, we turn to the diagnostic decomposition of the Boussinesq pressure equation (e.g., Markowski and Richardson, 2010), which may be written

$$c_{\rm p}\theta_{\rho 0}\nabla^2\pi' = \underbrace{-\nabla\cdot(\mathbf{u}\cdot\nabla)\mathbf{u}}_{\nabla^2 p'_{\rm d}} + \underbrace{\frac{\partial b}{\partial z}}_{\nabla^2 p'_{\rm b}}, \tag{3}$$

where $\theta_{\rho 0}$ is a reference value of $\theta_\rho$, $\mathbf{u}=(u,v,w)$, and $p'_{\rm b}$ and $p'_{\rm d}$ denote the buoyancy and dynamic pressure perturbation components, respectively. Away from solid boundaries, the above may be roughly simplified as

$$\pi' \propto -\nabla^2 p'_{\rm d} - \frac{\partial b}{\partial z}. \tag{4}$$

Neglecting the impacts of the $p'_{\rm d}$ term, larger vertical gradients in $b_{\rm co}$ are associated with larger adverse VPPG$_{\rm co}$, with lower $\pi'$ below the level of maximum buoyancy and higher $\pi'$ above it. This implies that the stronger VPPG$_{\rm co}$ sensitivity in the BOMEX

WIND and SCL-SHR experiments (relative to the corresponding ARM-SGP experiments) is, in part, associated with the larger $b_{\rm co}$ that develops at larger $U$. The cause of this wind-induced increase in $b_{\rm co}$ is examined in Sect. 4.2.

The sensitivities of $w_{\rm co}$ in the WIND and SCL-SHR experiments thus appear to be driven by variations in both the VPPG$_{\rm co}$ and $b_{\rm co}$ terms in Eq. (2). While VPPG$_{\rm co}$ exhibits a comparable decrease with increasing $U$ as in the CL-SHR experiments, offsetting variations in $b_{\rm co}$ lead to a muted sensitivity of $w_{\rm co}$ (Figs. 9e, f, i, j and Figs. 8c, e). The sensitivities of both $w_{\rm co}$ and $\varepsilon$





in these experiments are much stronger in BOMEX than in ARM-SGP. Unlike in the CL-SHR experiments, these differences
cannot be attributed to differential effects on vertical shear on cloud tilting because the cloud-layer shear is too weak. A physical
explanation for this behavior is thus required, and one will be provided in Sect. 4.2.

## 4.2   Properties of entrained air

The analysis in Sect. 4.1 showed that $w_{co}$ often correlates negatively with $\varepsilon$, which may be explained by the role of $w_{co}$ in

regulating the time scale over which ascending thermals are exposed to environmental air. However, the fact that $w_{co}$ did not
exclusively control $\varepsilon$ (Fig. 10a) suggests that other factors are needed to explain the $\varepsilon$ sensitivities. One such factor is the nature
of the entrained air in the core shell. As the fraction of environmental air within the shell decreases, it becomes less efficient
at diluting the cloud core, for all else being equal. Following Hannah (2017), we assume that entrained air is drawn from the
cloud-core shell. At each vertical level, this shell is defined following Dawe and Austin (2011) as all non-core grid points

immediately adjacent to core points, whether they are saturated or not. We further assume that core-shell air can be expressed
as a linear mixture of core and environmental air at that level. Any conserved variable (e.g., the total water specific humidity,
$s_t$) may thus be written

$$(s_t)_{sh} = \mu(s_t)_{co} + (1 - \mu)(s_t)_{en}, \tag{5}$$

where "en" and "sh" respectively denote the environment and cloud-core shell and $\mu$ is the mixing fraction, or the fraction

of cloud-core air within the cloud-core shell. Over all the simulations conducted herein, $\varepsilon$ tends to increase with $\mu^{-1}$, with a
correlation coefficient of $R = 0.84$ between them (Fig. 10b).

For the CTRL simulations, the cloud-layer-averaged $\mu$ is 0.53 for BOMEX and 0.70 for ARM-SGP (Fig. 11a). Thus, a given
entrainment flux yields less core dilution in ARM-SGP than in BOMEX. The imposed cloud-layer vertical wind shear in the
CL-SHR experiments has a negligible effect on $\mu$ in ARM-SGP, with a marginal increase of only 0.5 % between the CTRL and

S9 experiments. In contrast, BOMEX shows a more substantial 15 % corresponding increase (0.61), indicating a shift toward
less dilute cloud-core shells. Despite the stronger increase of $\mu$ in BOMEX across the CL-SHR experiments, $\mu$ is always larger
in ARM-SGP (Fig. 11a). This effect, along with the universally larger $w_{co}$ in ARM-SGP, largely explains why $\varepsilon$ is always
smaller in ARM-SGP than in BOMEX.

The BOMEX core shells also become less diluted as $U$ is increased in the WIND simulations, with $\mu$ increasing from 0.53

in CTRL to 0.60 in U10 (Fig. 11b). This increase is the largest between the CTRL and U2.5 cases and ultimately levels off
between the U5 and U10 cases. A similar lack of sensitivity between the U5 and U10 cases is apparent in the $w_{co}$, VPPG$_{co}$, $b_{co}$,
and $\varepsilon$ profiles (Figs. 8c, 9e-f, and 7a, respectively). As in the CL-SHR experiments, $\mu$ varies minimally in the corresponding
ARM-SGP experiments, with an increase of only 4 % from CTRL to U10. Whereas the addition of subcloud geostrophic shear
in the SCL-SHR experiments has a minimal additional impact on $\mu$ in ARM-SGP (Fig. 11c), it leads to a stronger and more

systematic increase in $\mu$ in BOMEX, to a value of 0.68 in US40. Although the variations in $\mu$ across the various suites of
simulations are modest, they suffice to explain the notable variations in $b_{co}$ in the BOMEX WIND and SCL-SHR experiments



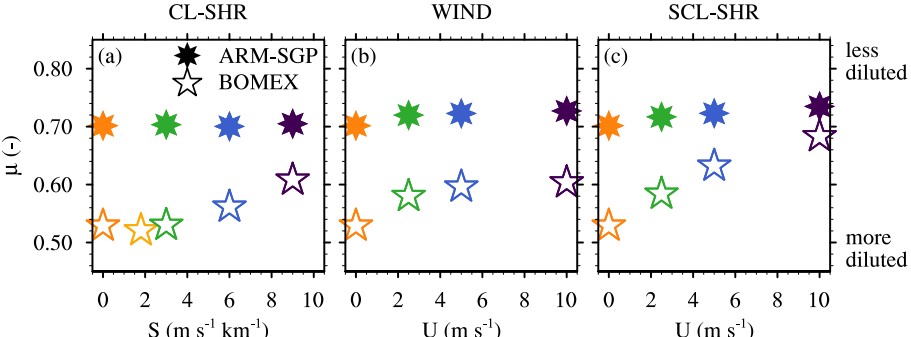

**Figure 11.** Mixing fraction $\mu$ for entraining nearest-neighbor adjacent core-exterior grid points for (**a**) the CL-SHR, (**b**) the WIND, and (**c**) the SCL-SHR experiments. The color scheme is identical to Fig. 9.

(Figs. 9b, f, and j). This is shown by a simple entraining parcel calculation that explicitly accounts for the cloud-core shell (Appendix B).

What controls the sensitivity of $\mu$ in BOMEX to the initial wind profile, and why is this sensitivity lacking in ARM-SGP?
These questions are addressed by looking farther afield than just the immediate core-adjacent grid points that constitute the cloud-core shell. To this end, we define a wider region surrounding the core as the "cloud-core margin", over which $\mu$ falls from its in-core values of approximately unity down to 0.5. This margin, which encapsulates the cloud-core shell, can be viewed as a finite-width halo of mixed air surrounding the core that shields it from pure environmental air. The width of this margin is henceforth denoted $R_{\mathrm{m}}$.

At each vertical level, we define $R_{\mathrm{m}}$ as the distance from the core edge to the nearest grid point where $\mu \leq 0.5$. This quantity is evaluated separately along both coordinate axes to compare the along- and cross-wind directions. The values of $R_{\mathrm{m}}$ thus presented are averaged over the central 50 % of the cloud layer (Fig. 12). While $R_{\mathrm{m}}$ is nearly axisymmetric for the CTRL cases (orange markers in Fig. 12), it develops anisotropy in the sensitivity tests. In the BOMEX CL-SHR experiments, $R_{\mathrm{m}}$ grows in all directions but to the largest degree (120 %) on the downshear side of the cloud cores (Fig. 12a). In ARM-SGP, $R_{\mathrm{m}}$ only
grows noticeably on the downshear side, by around 50 % (Fig. 12b). The downshear widening of $R_{\mathrm{m}}$ is consistent with the formation of a humid downshear wake (e.g., Heus and Jonker, 2008).

In the BOMEX WIND experiments, $R_{\mathrm{m}}$ again grows in all directions as $U$ is increased (Fig. 12c), but to a slightly lesser degree than in the corresponding CL-SHR experiments. Moreover, the core-margin expansion is maximized on the northern and western flanks of the cores, as opposed to the east side in the CL-SHR experiments, due to the weak east-southerly shear that develops in the lower cloud layer (Figs. 2a-b). For the ARM-SGP WIND experiments, $R_{\mathrm{m}}$ again undergoes less variation
than in the corresponding BOMEX experiments, with the largest expansion on the downshear (southeasterly) side of the cores (Fig. 12d). In the SCL-SHR experiments, $R_{\mathrm{m}}$ shows an even stronger sensitivity to $U$ in BOMEX while it remains virtually unchanged from the WIND experiments in ARM-SGP (Figs. 12e-f).

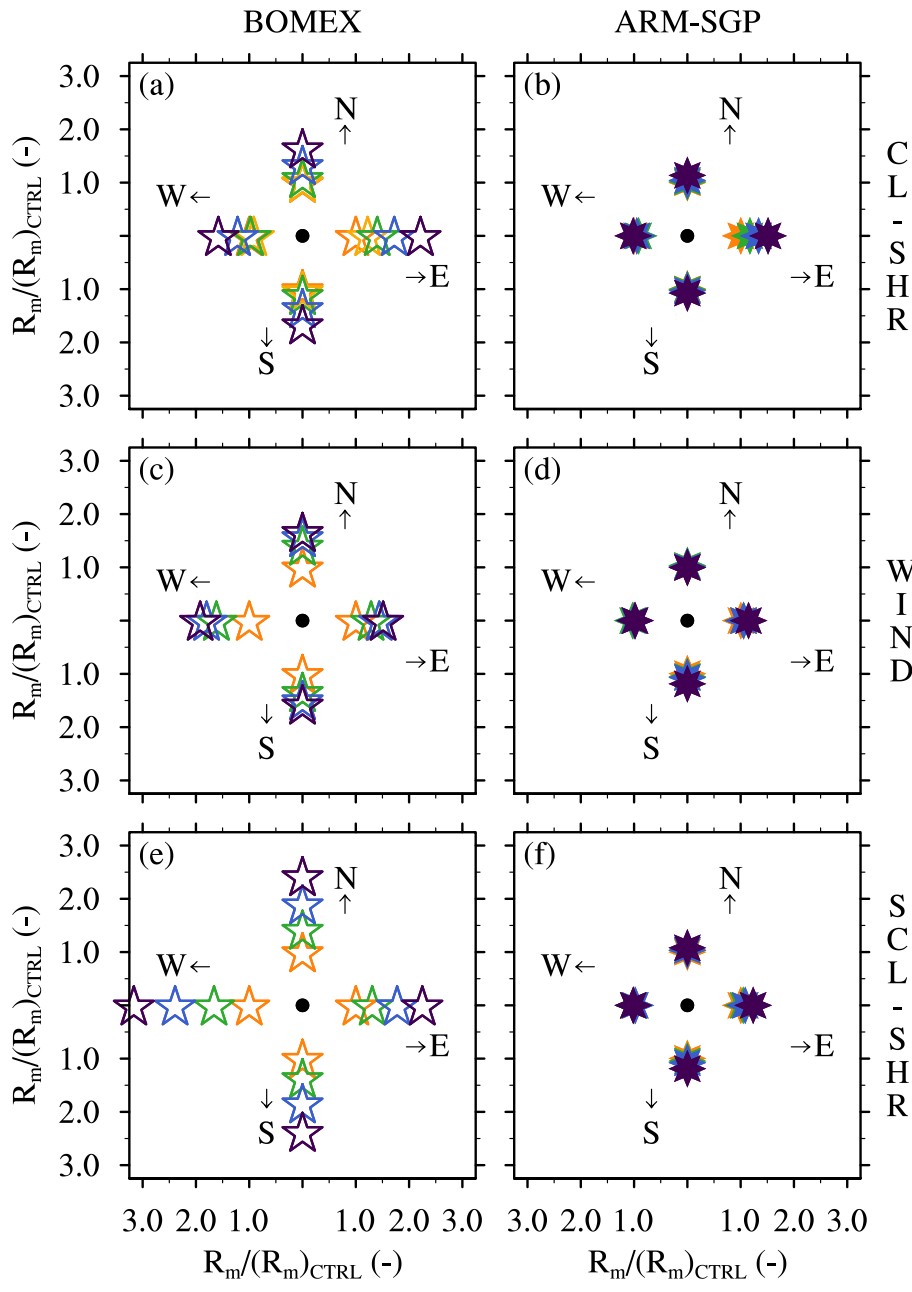

**Figure 12.** Radius of the cloud-core margin ($R_\mathrm{m}$) averaged over the central 50 % of cloud layer for the different directions normalized by the averaged $R_\mathrm{m}$ for the respective CTRL cases for (**a-b**) the CL-SHR, (**c-d**) the WIND, and (**e-f**) the SCL-SHR experiments. Panels (**a**), (**c**) and (**e**) show the BOMEX cases and panels (**b**), (**d**), and (**f**) the ARM-SGP cases. The color scheme is identical to Fig. 9.





In absolute terms, $R_\mathrm{m}$ averaged over all directions is about 60 % smaller in the CTRL BOMEX case than in the CTRL ARM-
SGP case (Figs. 13a-c). The cloud cores in BOMEX thus have narrower buffer zones surrounding them, which increases their
exposure to environmental air. Wider core margins tend to exhibit larger $\mu$ because, as the transition from core to environmental
air becomes more gradual, the air immediately adjacent to the core becomes more core-like. For the three sets of experiments,
the ARM-SGP $R_\mathrm{m}$ is generally less sensitive to changes in the wind profile than the corresponding BOMEX value. Whereas
the former exhibits a maximum increase of 17 % for the CL-SHR experiments, BOMEX exhibits a maximum increase of
153 % for the SCL-SHR experiments.

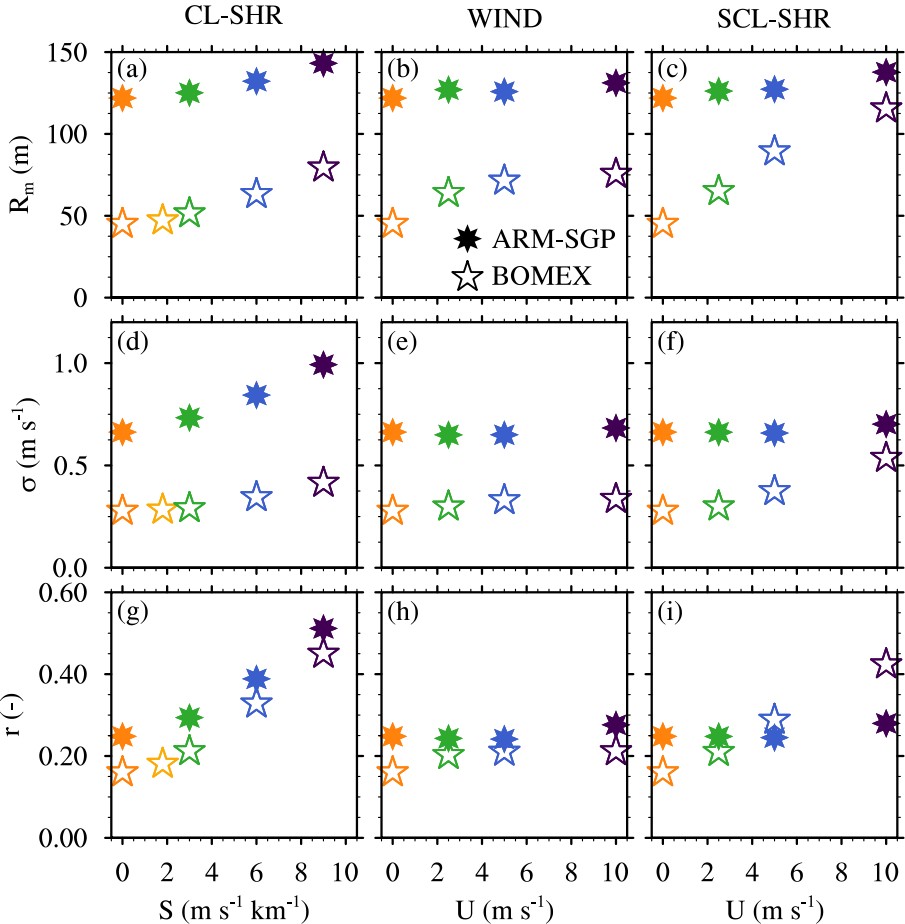

**Figure 13.** Various properties relevant to the (**a-c**) the cloud-core margin width averaged over all directions and the central 50 % of the cloud
layer ($R_\mathrm{m}$), (**d-f**) the square root of cloud-layer TKE ($\sigma$), and (**g-i**) $r = \sigma/w_\mathrm{co}$. Panels (**a**), (**d**), and (**g**) show the CL-SHR, panels (**b**), (**e**),
and (**h**) the WIND, and (**c**), (**f**), and (**i**) the SCL-SHR experiments. The color scheme is identical to Fig. 9.

In general, $R_\mathrm{m}$ correlates well with $\mu$ (cf. Figs. 13a-c and Fig. 11), with the lone exception being the ARM-SGP CL-SHR
experiments, where a 17 % enhancement in $R_\mathrm{m}$ does not coincide with increased $\mu$. We hypothesize that two factors combine



(Figs. 7c-d). In BOMEX SCL-SHR, larger increases in subcloud TKE and $\mathrm{TKE_{CL}}$ lead to even larger increases in $\mu$. However, the stronger corresponding reduction in $w_{\mathrm{co}}$ counters this effect to yield a similar $\varepsilon$ trend as that in BOMEX WIND.

Returning to the CL-SHR experiments, the positive sensitivity of $\varepsilon$ to vertical wind shear differs from Lin (1999), Brown (1999), and Helfer et al. (2020), who all found minimal corresponding sensitivities. These differences may be explained by a combination of factors. Because Lin (1999) evaluated $\varepsilon$ based on the vertical mass flux profile alone, they neglected the important role of detrainment in shaping that profile. Although Brown (1999) calculated $\varepsilon$ using a rigorous method (SC95), they used geostrophic shear profiles extending over both the subcloud and cloud layers. Given that cloud-layer shear and

subcloud shear have opposing effects on $\varepsilon$, it is possible that these two effects largely cancelled. Similar to Brown (1999), Helfer et al. (2020) used vertically constant shear profiles in their LES study. Furthermore, they employed the simpler "bulk-plume" method to $\varepsilon$, which neglects two of the terms in the SC95 formulation (Betts, 1975). More difficult to reconcile is the recent observational finding from Kirshbaum and Lamer (2021) that retrieved $\varepsilon$ does not vary systematically with cloud-layer shear, in oceanic or continental locations. It is possible that offsetting effects between subcloud and cloud-layer shear also

occur in reality, and/or that the differences between geostrophic winds (used herein) and full winds (used in Kirshbaum and Lamer, 2021) could explain these differences.

### 4.3 Empirical Relationship

Following from the results presented above, we have developed an empirical relationship for $\varepsilon$ that takes the two key controls on cloud dilution identified herein into account. These controls are the "core exposure effect" regulated by $w_{\mathrm{co}}$ and the "core-shell

dilution effect" (i.e. the amount of dilution per unit of entrainment) determined by $\mu$. As seen in Fig. 10, these two quantities vary roughly inversely with $\varepsilon$, which guides the form of the empirical function. Based on all the experiments conducted herein, we propose the following empirical function:

$$\varepsilon_{\mathrm{DKK}} = w_{\mathrm{co}}^{\alpha} \mu^{\beta} + \gamma, \tag{6}$$

with $\alpha = -1.14$, $\beta = -1.84$, and $\gamma = -0.2$. Calculated for CL-SHR, WIND, and SCL-SHR experiments, $\varepsilon_{\mathrm{DKK}}$ approximates

the simulated $\varepsilon$ very well (R=0.99; Fig. 15).

### 5    Conclusions

In this second part of a two-part study on the environmental controls on shallow-cumulus dilution, the impacts of variations in the geostrophic wind profile on cloud dilution have been investigated. To this end, LES experiments were conducted that systematically varied the cloud-layer vertical shear (CL-SHR; from 0 to 9 m s$^{-1}$ km$^{-1}$), the background wind speed (WIND;

from 0 to 10 m s$^{-1}$), and the subcloud (0-250 m above ground level) vertical shear (SCL-SHR; from 0 to 40 m s$^{-1}$ km$^{-1}$). To consider different shallow-cumulus manifestations observed in reality, these tests were run on both a quasi-statistically steady maritime, trade-wind flow (BOMEX) and a diurnally forced continental flow (ARM-SGP).

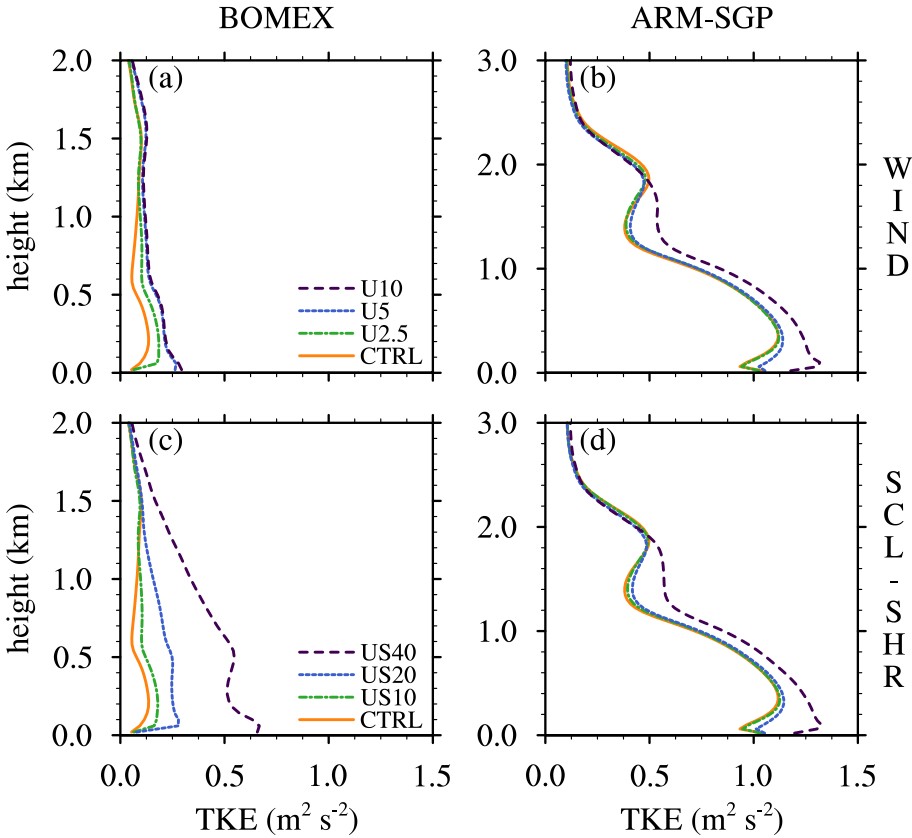

**Figure 14.** Vertical profiles of domain-averaged TKE for (**a-b**) the WIND and (**c-d**) the SCL-SHR experiments. Panels (**a**) and (**c**) show the BOMEX cases and the panels (**b**) and (**d**) the ARM-SGP cases.

Altogether, the experiments suggested that two basic factors control the sensitivity of the simulated cloud-core dilution rate ($\varepsilon$) to the imposed winds: the time scale over which the ascending cloud cores are exposed to environmental air, and the
mixing fraction ($\mu$, representing the fraction of cloud-core air within the mixture) of the "shell" immediately outside to the core, from which entrained air is drawn. The first effect, which we call the "core exposure effect", is directly controlled by the cloud-core vertical velocity $w_{co}$ and induces an inverse relationship between $w_{co}$ and $\varepsilon$ (e.g., Neggers et al., 2002). The second effect, called the "core-shell dilution effect", is largely controlled by the width of the buffer zone between core and environmental air. Larger widths exhibit more gradual transitions from core to environmental air, which give larger $\mu$ in the
grid points immediately adjacent to the core. These widths were largely controlled by the ratio of the square root of core-layer TKE to $w_{co}$.

The core-exposure and core-shell-dilution effects both depend inversely on $w_{co}$ and tend to mutually offset. For example, a decrease in $w_{co}$ increases the core-exposure time scale, which tends to enhance dilution, while also increasing the core-shell-mixing time scale, which tends to weaken dilution by increasing $\mu$. In the CL-SHR experiments, the vertical shear induced a





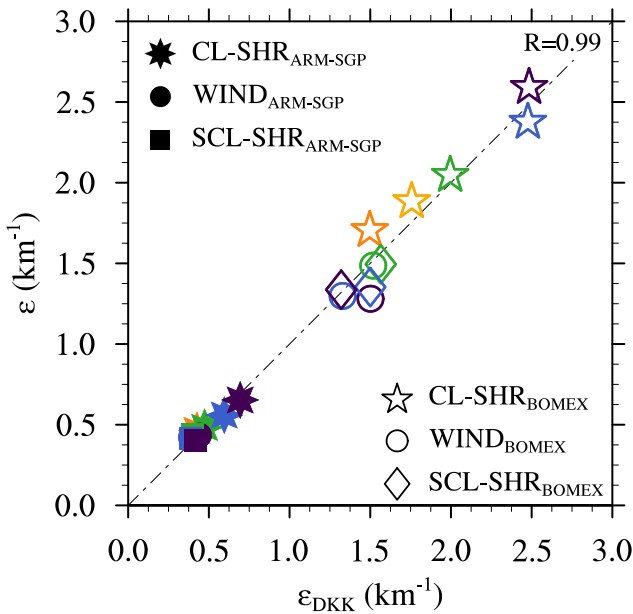

**Figure 15.** Relation of simulated dilution rate ($\varepsilon$) and dilution rates ($\varepsilon_{\mathrm{DKK}}$) obtained using the empirical formulation in Eq. (6).

large (up to 50 %) decrease in $w_{\mathrm{co}}$, owing to enhanced vertical perturbation pressure gradients suppressing the updrafts. As a result, the core-exposure effect tended to enhance dilution while the core-shell dilution effect tended to weaken it. In this case, the core-exposure effect dominated, leading to an increase in $\varepsilon$ (by up to 50 %) under stronger cloud-layer vertical shear.

In contrast, for the WIND and SCL-SHR experiments, the main sensitivities of $\varepsilon$ were traced to subcloud, rather than cloud-layer, processes. The strong near-surface shears in both cases (either prescribed or induced by surface drag) increased the

subcloud TKE, which extended into the cloud layer. As a result, the mixing rate within the cloud shells increased to give larger $\mu$, which favored a purification of the cloud cores. Although $w_{\mathrm{co}}$ also exhibited a small decrease with increasing winds, thereby activating the core-exposure effect, the core-shell-dilution effect was dominant, leading to decreased $\varepsilon$ (by up to 25 %) under increasing geostrophic winds (and subcloud shears). Thus, the effect of vertical shear on $\varepsilon$ depends on the layer where the shear is applied; cloud-layer shear enhances cloud dilution while subcloud shear decreases it.

The maritime BOMEX simulations were generally more sensitive to changes in the geostrophic wind profile than the ARM-SGP simulations, for two main reasons. Firstly, as found in the CL-SHR experiments, the weaker sensible heating over the ocean supports shallower subcloud layers with smaller-scale subcloud updrafts, which, in turn, initiate smaller cumuli. These cumuli were more susceptible to shear-induced tilting, and thus were more suppressed by the shear than the wider cumuli in ARM-SGP. Secondly, as found in the WIND and SCL-SHR experiments, the subcloud TKE was more sensitive to low-level

shear in BOMEX than in ARM-SGP, mainly because the weaker surface heating in BOMEX yielded a lower baseline TKE. Extension of this enhanced TKE into the cloud layer widened the transition zones between the cores and their environment,





thus inducing a purifying effect. Because the low-level TKE was only marginally enhanced by the subcloud shear in the corresponding ARM-SGP simulations, these cases were nearly insensitive to changes in the geostrophic wind profile.

The robust positive sensitivity of $\varepsilon$ to the cloud-layer shear in the CL-SHR differs from the findings of previous LES studies
(Lin, 1999; Brown, 1999; Helfer et al., 2020) and observational $\varepsilon$ retrievals (Kirshbaum and Lamer, 2021). While the former discrepancies can be explained by key differences in model initialization or $\varepsilon$ diagnoses, the latter is more concerning and merits future investigation. Such analysis would need to include the use of instrument simulators to ensure that both observed and simulated $\varepsilon$ are calculated for comparable subsets of shallow cumuli and the comparison is not compromised by the difficulty of observationally detecting clouds with small LWC. In contrast, the weakening of $\varepsilon$ with increasing background winds in
the BOMEX WIND and SCL-SHR is consistent with Kirshbaum and Lamer (2021), who found a robust inverse relationship between $U$ and $\varepsilon$ in the oceanic Eastern North Atlantic ARM site in the Azores. In a followup study, it would be interesting to investigate the cloud-core margin ($R_{\rm m}$) in observations and whether it can be related to reflectivity variability at each level within the cloud.

*Code and data availability.* The Bryan Cloud Model (CM1) is available under http://www2.mmm.ucar.edu/people/bryan/cm1/ (last access:
10 September 2020). Simulated data and analysis scripts as well as other supplementary information that may be useful for reproducing the author's work are archived by the Department of Atmospheric and Oceanic Sciences (McGill University) under https://aos.meteo.mcgill.ca/ (last access: 10 September 2020). The username and password can be obtained by contacting sonja.drueke@mail.mcgill.ca.

## Appendix A: Linear theory of shallow convection

Kirshbaum and Straub (2019) have used the linear theory for statically unstable cloud layers with background vertical wind
shear to examine the impact of vertical wind shear on shallow convection. This model is used to help interpret the stronger shear-induced suppression of cumuli in BOMEX than in ARM-SGP. In the linear theory, the convective growth rate ($\sigma$) is evaluated as a function of the nondimensional horizontal wavenumber $\kappa H$, where $\kappa = 2\pi/\lambda$ is the 2D horizontal wavenumber and $H$ is the depth of moist-unstable cloud layer, the latter characterized by negative Brunt-Väisälä frequency ($N_m^2$; Durran and Klemp, 1982).
To determine the applicability of the linear theory to our simulations, we compare the linear-predicted updraft suppression between the CTRL and S9 simulations for both BOMEX and ARM-SGP. For this analysis, $\kappa$ is assigned as the wavenumber of the spectral peak of the cloud-layer-averaged Fourier kinetic energy spectrum, $H$ is the depth of the layer over which $N_m^2 < 0$ (assuming saturated flow), and $N_m^2$ is averaged over $H$. A comparison of these quantities for the CTRL cases indicates smaller $\kappa$ (and hence larger horizontal scales) and a shallower unstable layer depth for ARM-SGP (Table A1), yielding smaller cloud
aspect ratios. Substituting these values, along with the zonal vertical shear magnitude, into the linear model of Kirshbaum and Straub (2019), we obtain the $\sigma$ values in Table A1.



**Table A1.** Summary of linear theory analysis. All symbols are defined in the text.

| | $\kappa$ (km$^{-1}$) | $H$ (km) | $N_{\mathrm{m}}^2$ ($10^{-5}$ s$^{-2}$) | $\sigma$ ($10^{-2}$ s$^{-1}$) |
|---|---|---|---|---|
| BOMEX-CTRL | 0.74 | 0.8 | -5.0 | 0.64 |
| BOMEX-S9 | 0.74 | 0.8 | -5.0 | 0.21 |
| ARM-SGP-CTRL | 0.44 | 0.6 | -6.0 | 0.58 |
| ARM-SGP-S9 | 0.44 | 0.6 | -6.0 | 0.26 |

## Appendix B: Parcel model

To show that the modest changes in $\mu$ across the WIND simulations (Fig. 11) suffice to explain the corresponding variations
in $b_{\mathrm{co}}$ in Fig. 9, we use a simple entraining parcel model similar to that developed by Hannah (2017) to illustrate the effect
of increased $\mu$ and $b_{\mathrm{co}}$. This model draws a mean-layer (0-500 m) parcel from the initial BOMEX sounding and adiabatically
lifts it to the base of the trade-wind inversion at 1.5 km. Above the LFC, it ingests surrounding air at a fixed rate of $\varepsilon_{\mathrm{p}}$, where
"p" denotes the parcel. Rather than entraining pure environmental air, the parcel entrains a mixture of core and environmental
air from the core shell. Assuming a statistically steady cloud field, and that the parcel equivalent potential temperature ($\theta_e$) is
conserved with height except for this mixing, the dilution may be estimated using

$$\frac{\partial (\theta_e)_{\mathrm{p}}}{\partial z} = -\varepsilon_{\mathrm{p}} \left( (\theta_e)_{\mathrm{p}} - (\theta_e)_{\mathrm{sh}} \right). \tag{B1}$$

The shell properties are related to those of the environment and parcel by $\mu$:

$$(\theta_e)_{\mathrm{sh}} = \mu (\theta_e)_{\mathrm{p}} + (1 - \mu)(\theta_e)_{\mathrm{en}}. \tag{B2}$$

Combining Eq. (B1) and Eq. (B2), we obtain

$$\frac{\partial (\theta_e)_{\mathrm{p}}}{\partial z} = -\varepsilon_{\mathrm{p}} (1 - \mu) \left( (\theta_e)_{\mathrm{p}} - (\theta_e)_{\mathrm{en}} \right). \tag{B3}$$

We solve Eq. (B3) numerically to obtain $(\theta_e)_{\mathrm{p}}$, and retrieve the parcel properties from it to evaluate $b_{\mathrm{p}}$.

The factor $\varepsilon_{\mathrm{p}}(1 - \mu)$ in Eq. (B3) indicates that, for $\mu > 0$, the core shell effectively weakens the cloud dilution from a given
entrainment rate $\varepsilon_{\mathrm{p}}$, and the strength of this effect increases with $\varepsilon_{\mathrm{p}}$. Because the $\varepsilon$ formulation in SC95 does not explicitly
account for the impacts of the core shell, $\varepsilon_p$ must exceed the SC95-calculated $\varepsilon$ to realize the same amount of core dilution.
Given that $\varepsilon \approx 1.5$ km$^{-1}$ and $\mu \approx 0.6$ in BOMEX CTRL, we set $\varepsilon_{\mathrm{p}} = 2.5$ km$^{-1}$ to yield similar cloud dilution as that in the
BOMEX simulations, thus facilitating a more direct comparison. This enhanced value of $\varepsilon_{\mathrm{p}}$ is similar in magnitude to the
LES-based direct entrainment rates reported in the literature (e.g, Romps, 2010; Dawe and Austin, 2011).

Figure B1 compares the parcel-model-derived $b_{\mathrm{p}}$ for the BOMEX case for $\mu = 0.53$ and $\mu = 0.60$, matching the range found
across the WIND simulations. For the chosen $\varepsilon_{\mathrm{p}}$, the magnitudes and sensitivities $b_{\mathrm{p}}$ are very similar, if not larger, to those



found in the corresponding BOMEX WIND simulations (Fig. 9f). Thus, the variations in $b_{co}$ in the BOMEX WIND and SCL-
SHR sensitivity tests can largely be explained by corresponding variations in $\mu$. Similarly, the minimal variations in $b_{co}$ among
the corresponding ARM-SGP experiments are consistent with their minimal $\mu$ sensitivities. This analysis does not carry over
to the CL-SHR experiments because the variations in $\mu$ coincide with large variations in $w_{co}$, which may also impact $b_{co}$.

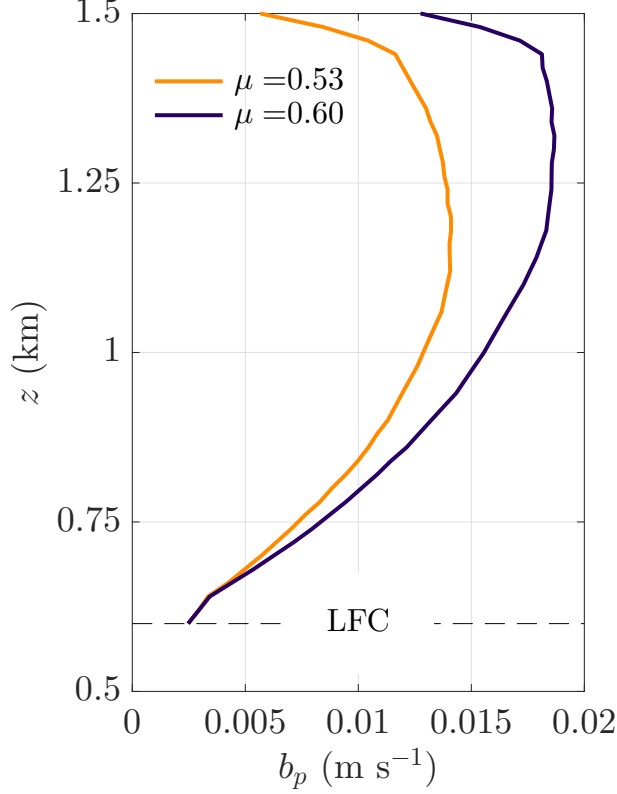

**Figure B1.** Sensitivity of entraining-parcel-model buoyancy ($b_{\mathrm{p}}$) to core-shell mixing fraction ($\mu$) for the initial BOMEX sounding, assuming
a vertically constant shell-entrainment rate of $\varepsilon_{\mathrm{p}} = 2.5 \ \mathrm{km}^{-1}$.

*Author contributions.* S.D. and D.J.K. developed the scientific question, and S.D. conducted the simulations and carried out the analysis
under the supervision of D.J.K. and co-supervision of P.K. S.D. prepared the paper with contributions from D.J.K. and P.K.

*Competing interests.* The authors declare that they have no conflict of interest.



*Acknowledgements.* Research funding was provided from the Natural Sciences and Engineering Research Council (NSERC) Grant NSERC/ RGPIN 418372-17 and the US Department of Energy Atmospheric System Research (DOE–ASR) Program under contract DE-SC0020083. P.K. was supported by the US Department of Energy (DOE) Atmospheric System Research Program under contract DE-SC0012704. The numerical simulations were performed on the Guillimin supercomputer at McGill University and Béluga supercomputer at the École de technologie supérieure, both under the auspices of Calcul Québec and Compute Canada.




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
