# Peer review of "Environmental sensitivities of shallow-cumulus dilution – Part 2: Vertical wind profile"

_Atmospheric Chemistry and Physics, 2021_

## Referee Comment (RC1)

**Environmental sensitivities of shallow-cumulus dilution**
**Part 2: Vertical wind profile**

*S. Drueke, D. J. Kirshbaum, and P. Kollias*

This study explores how cloud dilution is affected by environmental shear using simulation ensembles of shallow convection in maritime and continental conditions. Idealized shear profiles are designed to test the effects of shear within the cloud and subcloud layers separately. These experiments produce contrasting results in that cloud layer shear leads to enhanced dilution, while subcloud layer shear leads to reduced dilution. The authors then use theory and simple calculations to discuss the various factors driving these behaviors.

I really like this paper. The experiments and analysis are thoughtfully designed and clearly presented. The writing is also very clear. I think it will make a great contribution to the existing literature.

On a personal note, I've tried to investigate this same question of how shear influences dilution in several different ways, but I never found any notable relationship. After reading this manuscript I realize that all my experiments were similar to the ARM SGP simulations presented here in that the convection was mainly forced by a large buoyancy source, and my shear profiles created a cancellation between competing effects. So I really appreciate that the experiments here demonstrate how the effects of shear are conditional on how the convection is being forced and whether the shear is in or below the cloud layer.

The language tends to focus on how the experiments are looking to make a direct connection between shear and dilution, but the results seem to confirm that the changes to dilution are secondary to how the shear impacts the cloud momentum and initial turbulence statistics. This doesn't make the authors' wording right or wrong, but I do wonder if the summary language (maybe the abstract as well) could be altered to reflect this. This is a pretty loose observation, so take what you want from it.

I have outlined a few other minor concerns below, but otherwise I think the paper is in great shape.

**Are the WIND experiments really needed?**
I'm not convinced that the "uniform wind" experiments add much to the analysis given that the surface friction leads to subcloud layer shear that is so similar to the SCL-SHR experiments. If the wind was nudged to maintain a profile that was closer to the initial profile then I would feel differently, but as it stands the data from the WIND and SCL-SHR results are very similar, so the WIND experiments don't seem to add much insight.

It also seems sensible to show the CL-SHR and SCL-SHR results next to each other to better highlight the contrast in the dilution results. For instance, combining Figure 5a-b with

Figure 7c-d would highlight the main conclusion more clearly in my opinion. I realize this might involve a substantial reorganization of the paper, but it might be worth it. I'll leave it to the author to decide whether to keep the current form or not since the main conclusion is not affected either way.

**Connection between shear and updraft width**
Since this paper talks about cloud width in sheared environments it might be good to mention the results of this recent paper by John Peters:

Peters, J. M., Nowotarski, C. J., & Morrison, H. (2019). The Role of Vertical Wind Shear in Modulating Maximum Supercell Updraft Velocities, *J. of the Atmos. Sci.*, **76**(10), 3169-3189.

That paper is not directly applicable since it focuses on supercell storms, but the connection they draw between cloud width and storm relative inlow seems like it could play a role in other cloud types. The relationship between subcloud layer TKE and cloud width in the simulations presented here is probably the more accurate and relevant explanation for shallow convection, but the consistency between these results that more shear always leads to wider clouds is intriguing.

**Dilution estimation terminology and method**
On line 189 the authors introduce the dilution calculation method by saying:

"*Diagnosis of the simulated bulk fractional entrainment rate, or simply the 'dilution rate'...*"

I think it's important to highlight that the "bulk dilution rate" calculated here includes the effects of both entrainment and detrainment (at least that's what I get from the description of the calculation). I know that often "bulk entrainment" implies that the detrainment is assumed to be zero (except at "cloud top"), but there's been so much inconsistent use of this terminology that it seems important to be as explicit as possible when discussing this stuff. I would suggest doing a little rewording to clarify what the bulk dilution estimate is actually measuring, and note that it's not just the effects of entrainment that would be estimated by a direct measurement scheme, like the ones by David Romps and others.

It also might be worth showing the budget equation used for the bulk dilution calculation. It might seem unnecessary since budget equations like this have been shown in so many papers, but I think it helps the reader understand the nuances of what is actually being calculated when the equations are shown in their full detail. This wouldn't need to include any sort of derivation, just the final equation that the dilution estimate is based on.

**Use of the sigma symbol**
I got thrown off a little bit due to the sigma symbol being used for 2 different things: the convective growth rate on line 295 and the cloud layer TKE on line 395. I think it's worth changing one of them for clarity's sake.

**Aerosols**

I'm not normally someone to bring this up, but since you're comparing continental and maritime environments it would be good to mention if there's a difference in CCN concentrations. I couldn't find a mention of this in the current manuscript, so let me know if I missed it. I doubt the droplet size distribution would matter for this study, but it's been such a hot topic in the field that it's always good to mention how the model is configured in this respect.

- Walter Hannah

---

## Author Comment (AC1)

**Responses to referee comments, Atmos. Chem. Phys. Discuss. acp-2020-336**

We are very grateful to the two reviewers for their insightful and constructive comments, which have helped to substantially improve our analysis and physical interpretation. In the following, reviewer comments are written in black, author responses are written in blue, and passages of modified text are written in red.

**Referee 1 – Walter Hannah**

General comments: This study explores how cloud dilution is affected by environmental shear using simulation ensembles of shallow convection in maritime and continental conditions. Idealized shear profiles are designed to test the effects of shear within the cloud and subcloud layers separately. These experiments produce contrasting results in that cloud layer shear leads to enhanced dilution, while subcloud layer shear leads to reduced dilution. The authors then use theory and simple calculations to discuss the various factors driving these behaviors.

I really like this paper. The experiments and analysis are thoughtfully designed and clearly presented. The writing is also very clear. I think it will make a great contribution to the existing literature.

On a personal note, I've tried to investigate this same question of how shear influences dilution in several different ways, but I never found any notable relationship. After reading this manuscript I realize that all my experiments were similar to the ARM SGP simulations presented here in that the convection was mainly forced by a large buoyancy source, and my shear profiles created a cancellation between competing effects. So I really appreciate that the experiments here demonstrate how the effects of shear are conditional on how the convection is being forced and whether the shear is in or below the cloud layer.

The language tends to focus on how the experiments are looking to make a direct connection between shear and dilution, but the results seem to confirm that the changes to dilution are secondary to how the shear impacts the cloud momentum and initial turbulence statistics. This doesn't make the authors' wording right or wrong, but I do wonder if the summary language (maybe the abstract as well) could be altered to reflect this. This is a pretty loose observation, so take what you want from it.

I have outlined a few other minor concerns below, but otherwise I think the paper is in great shape.

We thank Walter for this positive review and his insightful and valuable comments.

**1. Are the WIND experiments really needed?**

I'm not convinced that the "uniform wind" experiments add much to the analysis given that the surface friction leads to subcloud layer shear that is so similar to the SCL-SHR experiments. If the wind was nudged to maintain a profile that was closer to the initial profile then I would feel differently, but as it stands the data from the WIND and SCL-SHR results are very similar, so the WIND experiments don't seem to add much insight.

It also seems sensible to show the CL-SHR and SCL-SHR results next to each other to better highlight the contrast in the dilution results. For instance, combining Figure 5a-b with Figure 7c-d would highlight the main conclusion more clearly in my opinion. I realize this might involve a substantial reorganization of the paper, but it might be worth it. I'll leave it to the author to decide whether to keep the current form or not since the main conclusion is not affected either way.

We understand the reviewer's concern regarding the WIND experiments as both the WIND and SCL-SHR experiments have similar findings, resulting from similar physical mechanisms. However, we believe that the WIND experiments are beneficial as they show the robustness of the results regardless of the source of the subcloud wind shear. Namely, they show that both low-level geostrophic shear **and** frictionally induced surface drag have similar effects on cloud-layer dilution. If we had only shown the former, readers may not have realized the implications of the findings. Because surface drag is universal, it effects all flows regardless of the level of geostrophic shear. Thus, by showing the WIND cases, the impacts of our findings are potentially enhanced. Also, it is worth noting that, although WIND and SCL-SHR have features in common, they are not identical. The deeper and stronger initial shear in SCL-SHR leads to a deeper extension of the shear into the cloud layer, which, in turn, enhances dilution near cloud base. This effect is not seen in the WIND cases, so the comparison of these two experiments helps to isolate its underlying cause. Hence, we have decided to keep the WIND experiments as a part of the manuscript. Nevertheless, we appreciate that a more detailed justification for the WIND experiments is needed. Thus, the following passage was added to the manuscript in lls. 265-270:

Although the WIND and SCL-SHR experiments reach a common conclusion that subcloud-layer shear tends to decrease cloud-layer dilution, the consideration of both sets of simulations aids physical interpretation. For one thing, it shows that geostrophic shear is not required to realize this trend; even frictionally induced shear at the surface, which is present in all flows, suffices. Also, as mentioned above, the SCL-SHR experiments reveal an effect that was absent in WIND: a systematic enhancement in lower-cloud-layer dilution. Attribution of this effect to the enhanced lower-cloud-layer shear is facilitated by the WIND experiments, where the lower-cloud-layer shear and dilution vary much less between the different cases.

**2. Connection between shear and updraft width**

Since this paper talks about cloud width in sheared environments it might be good to mention the results of this recent paper by John Peters:

Peters, J. M., Nowotarski, C. J., & Morrison, H. (2019). The Role of Vertical Wind Shear in Modulating Maximum Supercell Updraft Velocities, J. of the Atmos. Sci., 76(10), 3169-3189.

That paper is not directly applicable since it focuses on supercell storms, but the connection they draw between cloud width and storm relative inflow seems like it could play a role in other cloud types. The relationship between subcloud layer TKE and cloud width in the simulations presented here is probably the more accurate and relevant explanation for shallow convection, but the consistency between these results that more shear always leads to wider clouds is intriguing.

Thank you for the suggestion. We have included a short passage of Peters et al.'s paper to our manuscript in lls. 91-92 as well a reference in lls. 236-237

For the special case of supercells, Peters et al. (2019) found that stronger vertical wind shear may indirectly weaken cloud dilution by enhancing cloud inflow and cell width.

**3. Dilution estimation terminology and method**

On line 189 the authors introduce the dilution calculation method by saying: "Diagnosis of the simulated bulk fractional entrainment rate, or simply the 'dilution rate'..."

I think it's important to highlight that the "bulk dilution rate" calculated here includes the effects of both entrainment and detrainment (at least that's what I get from the description of the calculation). I know that often "bulk entrainment" implies that the detrainment is assumed to be zero (except at "cloud top"), but there's been so much inconsistent use of this terminology that it seems important to be as explicit as possible when discussing this stuff. I would suggest doing a little rewording to clarify what the bulk dilution estimate is actually measuring, and note that it's not just the effects of entrainment that would be estimated by a direct measurement scheme, like the ones by David Romps and others.

We agree with the reviewer the terminology surrounding entrainment and dilution is at times used inconsistently, which leads to confusion. The bulk entrainment rate calculation used in this manuscript, which follows the formulation by Siebesma and Cuijpers (1995), is a measure of how diluted the cloud cores are as a consequence of mixing with pure environmental air. Its entrainment evaluation depends largely on a conserved scalar quantity (here  $s_t$ ), and the detailed mixing processes across the cloud perimeter (i.e. direct entrainment and detrainment) are not assessed. To clarify our dilution calculation, we have added the following sentence to the description of the dilution calculation in lls 193-195:

This quantity measures how much pure environmental would need to be entrained to achieve the simulated core dilution. Contrary to direct entrainment calculations, it does not quantify the actual mixing across the cloud perimeter.

It also might be worth showing the budget equation used for the bulk dilution calculation. It might seem unnecessary since budget equations like this have been shown in so many papers, but I think it helps the reader understand the nuances of what is actually being calculated when the equations are shown in their full detail. This wouldn't need to include any sort of derivation, just the final equation that the dilution estimate is based on.

Thank you for the suggestion, we have added the equation used for the dilution calculation in lls. 196-203.

**4. Use of the sigma symbol**

I got thrown off a little bit due to the sigma symbol being used for 2 different things: the convective growth rate on line 295 and the cloud layer TKE on line 395. I think it's worth changing one of them for clarity's sake.

We agree with the reviewer, the double use of the sigma symbol is unfortunate. While we have kept the use of sigma for the convective growth rate, we have have changed the second use of sigma. The cloud-layer TKE is henceforth indicated more straightforwardly by  $\text{TKE}_{cl}^{1/2}$ .

**5. Aerosols**

I'm not normally someone to bring this up, but since you're comparing continental and maritime environments it would be good to mention if there's a difference in CCN concentrations. I couldn't find a mention of this in the current manuscript, so let me know if I missed it. I doubt the droplet size distribution would matter for this study, but it's been such a hot topic in the field that it's always good to mention how the model is configured in this respect.

Thank you for the suggestion. We have added the difference in CCN concentrations to the description of the two environments in lls 163-164.

**Referee 2**

This is an excellent example of how to design and analyze a set of LES experiments to provide physical insight into the effect of shear on cloud-environment mixing in shallow convection. The authors use an initial wind profile that limits shear to the cloud layer, and show that shear-induced changes in both cloud-core updrafts and the character of the cloud shell control cloud dilution in these shallow clouds. The combination of modeling and simple theory are very well presented and the conclusion that cloud-core updraft velocity and the fraction of cloudy air in the cloud shell are controlling parameters is compelling and should will be of general interest.

We thank the reviewer for the very positive review and their time and the helpful comments.

**Minor comment:**

 Resolution: The distinction between dilution and entrainment is an important one, and I'm happy to see it called out here (although I agree with Walter Hannah that the discussion could be sharpened). I'm less certain about the use of "purifying effect" to describe the role the cloud shell plays in mediating mixing between cloud core and environment. On a two-point mixing line, environment and cloud core are equally "pure", and I think of purification as the removal of impurities – wouldn't "buffering" be a better fit? Not too invested in this however.

Thank you. We agree with the reviewer and have changed the text accordingly by substituting "buffering" for "purifying" in the description of the effect of the cloud shell.

**References**

- Peters, J. M., C. J. Nowotarski, and H. Morrison, 2019: The Role of Vertical Wind Shear in Modulating Maximum Supercell Updraft Velocities. J. Atmos. Sci., 76, 3169–3189, doi:10.1175/JAS-D-19-0096.1.
- Siebesma, A. P. and J. W. M. Cuijpers, 1995: Parametric Assumptions for Shallow Cumulus Convection. J. Atmos. Sci., 52 (6), 650–666, doi:10.1175/1520-0469(1995)052(0650:EOPAFS)2.0.CO;2.